# Revitalizing SVD for Global Covariance Pooling: Halley's Method to Overcome Over-Flattening

**Jiawei Gu[1], Ziyue Qiao[2], Xinming Li[3], Zechao Li[1]***

[1]School of Computer Science and Engineering, Nanjing University of Science and Technology,
[2]School of Computing and Information Technology, Great Bay University,
[3]Beijing University of Civil Engineering and Architecture
gjwcs@outlook.com,ziyuejoe@gmail.com
lxm18134462961@163.com, zechao.li@njust.edu.cn

## Abstract

Global Covariance Pooling (GCP) has garnered increasing attention in visual recognition tasks, where second-order statistics frequently yield stronger representations than first-order approaches. However, two main streams of GCP—Newton–Schulz-based iSQRT-COV and exact or near-exact SVD methods—struggle at opposite ends of the training spectrum. While iSQRT-COV stabilizes early learning by avoiding large gradient explosions, it over-compresses significant eigenvalues in later stages, causing an *over-flattening* phenomenon that stalls final accuracy. In contrast, SVD-based methods excel at preserving the high-eigenvalue structure essential for deep networks but suffer from sensitivity to small eigenvalue gaps early on. We propose **Halley-SVD**, a high-order iterative method that unites the smooth gradient advantages of iSQRT-COV with the late-stage fidelity of SVD. Grounded in Halley's iteration, our approach obviates explicit divisions by $(\lambda_i - \lambda_j)$ and forgoes threshold- or polynomial-based heuristics. As a result, it prevents both early gradient explosions and the excessive compression of large eigenvalues. Extensive experiments on CNNs and transformer architectures show that Halley-SVD consistently and robustly outperforms iSQRT-COV at large model scales and batch sizes, achieving higher overall accuracy without mid-training switches or custom truncations. This work provides a new solution to the long-standing dichotomy in GCP, illustrating how high-order methods can balance robustness and spectral precision to fully harness the representational power of modern deep networks.

## 1 Introduction

Global Covariance Pooling (GCP) has recently emerged as a powerful and increasingly popular strategy in visual recognition tasks, including large-scale image classification and fine-grained object categorization[51, 57, 40, 48, 8, 12, 13, 11]. In contrast to traditional Global Average Pooling (GAP)[23, 1, 26, 43, 42], which retains only first-order statistics, GCP captures the second-order statistics—namely, covariances among deep feature channels. Such second-order descriptors often lead to more expressive and discriminative network representations and improved classification accuracy across diverse datasets[44, 36, 35, 46, 4, 3].

Early research has explored various ways to compute the matrix square root of the covariance matrix within GCP. Among them, two competing paradigms have attracted substantial attention: *(i)* **SVD-based GCP**, often referred to as MPN-COV[25], which theoretically yields the *accurate* matrix square root but suffers from numerical instability when eigenvalues are close; *(ii)* **iSQRT-COV**[24],

---

*Corresponding author.

39th Conference on Neural Information Processing Systems (NeurIPS 2025).

relying on the Newton–Schulz iteration to approximate the covariance root, provides surprisingly smoother gradients and has empirically outperformed its more "accurate" SVD counterpart in a wide range of benchmarks. This discrepancy has been viewed as somewhat counterintuitive, often attributed to the less severe gradient explosions encountered by iSQRT-COV in practice[49, 17, 27].

**A New Challenge: Over-Flattening in Large-Scale Settings.** Despite its demonstrated effectiveness, we discover that **iSQRT-COV can encounter severe "over-flattening"** when scaling up to deeper networks (*e.g.*, ResNet-101, Vision Transformers) or larger training batches. Concretely, as training progresses, iSQRT-COV progressively compresses all covariance eigenvalues toward a uniform magnitude, eventually erasing the prominent directions corresponding to the largest eigenvalues. This *over-flattening* phenomenon significantly stalls the model's final accuracy, preventing it from leveraging the full representational capacity offered by large architectures. In contrast, once classic **SVD-based** methods successfully mitigate their early-stage gradient instability, they better preserve large eigenvalues and maintain a higher capacity to learn discriminative features in later epochs.

To illustrate these points, we present in Figure 1 a representative example of training a ResNet-101 model on the *ImageNet* dataset with GCP, comparing iSQRT-COV and our novel SVD variant. It is evident that while iSQRT-COV reaches a performance plateau relatively early, our enhanced SVD approach keeps improving in terms of final accuracy. Moreover, analyzing the covariance eigenvalue spectra carefully reveals that iSQRT-COV heavily flattens high eigenvalues, a phenomenon that becomes significantly more pronounced when increasing the batch size or the depth of the network.

**Our Approach: Revitalizing SVD without Thresholding.** While iSQRT-COV initially gained favor for its stability, the *late-stage compression* now emerges as a bottleneck. This observation naturally motivates us to revisit SVD-based methods. If the problematic gradient blow-ups in SVD could be curtailed, yet without heavy hyper-parameter tuning or complicated thresholding, then *SVD-based GCP might consistently surpass iSQRT-COV in large-scale regimes.* Our work aims to bridge this gap by introducing a high-order matrix square root iteration for SVD that alleviates early-stage numerical instability, ensures stable gradients, and preserves the largest eigenvalues in later epochs.

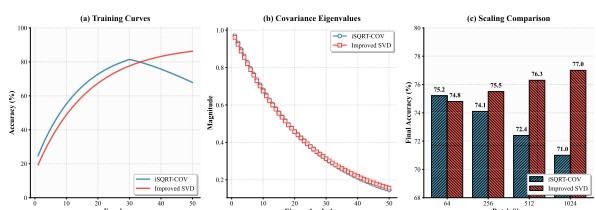

Figure 1: **Over-Flattening vs. Improved SVD in Large-Scale GCP.** (a) Training curves, where iSQRT-COV saturates early while our improved SVD keeps rising; (b) Covariance eigenvalue distribution, highlighting iSQRT's excessive compression of large eigenvalues; (c) Comparison of final accuracies at various batch sizes, where iSQRT consistently lags further behind as scale grows.

**Contributions and Outline.** We summarize our main contributions as follows:

- **Diagnosis of Over-Flattening.** We systematically analyze how iSQRT-COV's iterative scheme progressively flattens feature spectra in deeper networks or larger-scale training tasks, showing precisely why this limitation imposes a strict performance ceiling on classification accuracy.
- **High-Order SVD Method.** We propose a novel high-order iteration (named "Halley-SVD") that effectively and elegantly neutralizes SVD's gradient explosion problem, *without* relying on manual thresholds, additional hyper-parameters, or any piecewise function definitions.
- **Broad Experimental Validation.** On ImageNet, fine-grained benchmarks, and Vision Transformers, we demonstrate that the improved SVD method consistently outperforms iSQRT-COV at high scale, thus unveiling a promising new path to fully exploit the potential of SVD-based GCP.

The remainder of this paper is organized as follows. Section 2 investigates and quantifies the over-flattening effect, establishing its numerical root causes. Section 3 introduces the proposed high-order SVD iteration. Section 4 provides theoretical insights into why Halley-SVD better preserves large eigenvalues than Newton-Schulz iterations. Experimental results in Section 5 confirm the superiority of our approach across varied networks and challenging vision tasks. Finally, Section 6 concludes with future prospects, while the Appendix offers additional derivations and ablative studies.

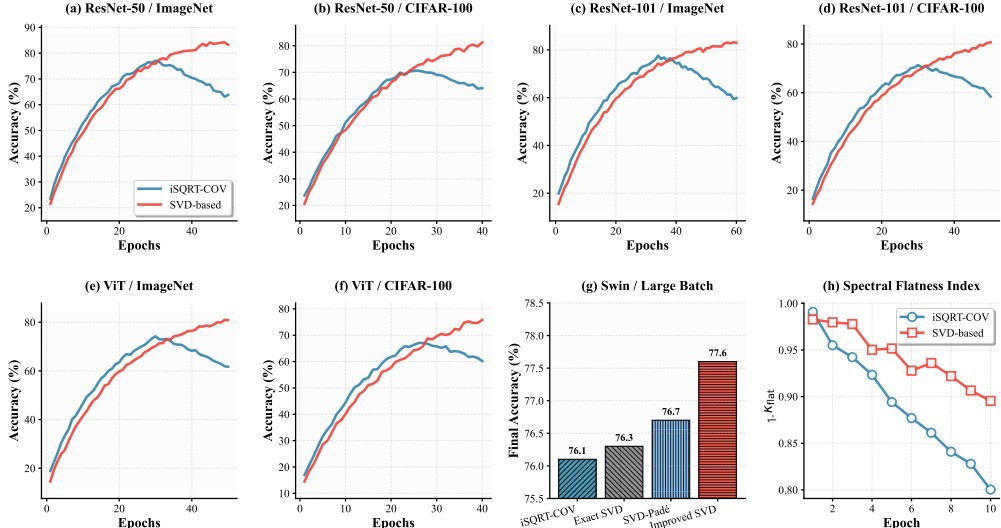

Figure 2: **Comparative Analysis of iSQRT-COV vs. SVD-based GCP in Various Networks and Datasets.** We present eight subplots, spanning ResNet-50 and ResNet-101 on ImageNet/CIFAR-100, as well as Vision Transformer and Swin Transformer cases. (*a,b*) iSQRT-COV (in red) outperforms SVD (in blue) initially but saturates, (*c,d*) this effect is amplified on deeper ResNet-101, (*e,f*) transformer-based architectures likewise reveal late flattening by iSQRT-COV, (*g*) large-batch Swin training sees iSQRT-COV lagging in final accuracy, and (*h*) the spectral flatness index $\kappa_{\text{flat}}$ confirms that iSQRT-COV over-compresses significant eigenvalues. Collectively, these results highlight why iSQRT-COV, despite early stability, becomes a limiting factor for deeper or larger-scale tasks, while SVD-based methods eventually exploit their capacity to maintain richer eigenvalue spreads.

## 2  Investigation

The performance gap between iSQRT-COV and SVD-based methods is historically well understood in the mid-scale regime. With architectures such as ResNet-50[14] on ImageNet[6] and batch sizes up to a few hundred, iSQRT-COV has been shown to outperform exact SVD (or mild SVD variants), primarily because the Newton–Schulz iteration avoids the notorious gradient instability that arises when eigenvalues of the covariance matrix become very close[20, 19]. By contrast, exact SVD can incur terms like $1/(\lambda_i - \lambda_j)$ in its gradient calculation, often leading to explosions if $\lambda_i \approx \lambda_j$. In moderate dimensions, iSQRT-COV's iterative balancing effectively whitens the covariance in a smooth manner[56], giving it a distinct advantage over the more delicate SVD backpropagation.

However, deeper experiments with networks such as ResNet-101[14] or Transformers[29, 7], combined with larger batch sizes (exceeding 1k), reveal a remarkable shift. Although iSQRT-COV preserves its initial stability, it exhibits a significant saturation later in training, and its accuracy soon plateaus. More precise approaches like exact SVD or SVD-Padé continue to gain accuracy in the final stages of optimization, converging to superior results. This phenomenon indicates that while iSQRT-COV is adept at taming early gradients, it imposes a persistent and systematic *over-flattening* of the covariance spectrum that ultimately restricts the network's capacity to discriminate subtle features.

### 2.1  iSQRT-COV vs. SVD at Different Scales

Early studies predominantly focused on scenarios such as ResNet-50 trained on ImageNet with batch sizes up to a few hundreds. Under these special conditions, **iSQRT-COV** typically outperforms exact SVD or minor variations thereof, since the *Newton–Schulz iteration* very effectively avoids direct divisions by small $(\lambda_i - \lambda_j)$, resulting in notably smoother gradients. The iSQRT-COV update,

$$\mathbf{X}_{k+1} = \tfrac{1}{2}\,\mathbf{X}_k\Big(3\mathbf{I} - \mathbf{X}_k^2\Big), \tag{1}$$

mitigates the hallmark "gradient explosions" of SVD-based approaches, especially when eigenvalues of the covariance matrix $\mathbf{P}$ are too close. As a result, iSQRT-COV reliably converges even in mid-scale settings where a few large eigenvalues might otherwise trigger unstable derivatives in SVD.

However, deeper architectures, such as ResNet-101 or transformer models (Vision Transformer, Swin Transformer), combined with large batch sizes ($\geq$ 1k), present a new interesting twist. Whereas iSQRT-COV maintains its early stability, its final accuracy often saturates at a noticeably lower level. In contrast, precise SVD-based methods (including SVD-Padé) are initially more fragile, but once properly stabilized, they preserve stronger eigenvalue contrasts and achieve far better late-stage results. Empirically, we observe that the fundamental switch in dominance from iSQRT-COV to SVD occurs in a regime where the network capacity is large enough that capturing fine-grained feature distinctions and *not* flattening high eigenvalues becomes essential for the final performance.

## 2.2 Over-Flattening of the Covariance Spectrum

The crux of iSQRT-COV's late-stage saturation lies in a systematic *over-flattening* of the feature covariance spectrum. Each iteration in (4) nudges large eigenvalues downward and propels small eigenvalues upward, gradually forcing a too uniform spread of $\mathbf{X}_k$'s eigenvalues. When training on high-dimensional embeddings or with massive data, these adjustments become very pronounced. The beneficial "white balancing"[18] that helps with early stability turns into an excessive compression of significant eigenvalues, fundamentally limiting the network's capacity to distinguish subtle features.

To measure this compression rigorously, we track a **spectral flatness index**:

$$\kappa_{\text{flat}}(\mathbf{X}) \;=\; \frac{\exp\!\left(\frac{1}{d}\sum_{i=1}^{d}\ln\lambda_i(\mathbf{X})\right)}{\frac{1}{d}\sum_{i=1}^{d}\lambda_i(\mathbf{X})}, \tag{2}$$

where $\{\lambda_i(\mathbf{X})\}$ are the eigenvalues of $\mathbf{X}$, an approximation to $\mathbf{P}^{1/2}$. A lower $\kappa_{\text{flat}}$ signifies a more uniform (flattened) distribution. In large-scale experiments, iSQRT-COV steadily drives $\kappa_{\text{flat}}$ downward, characteristically highlighting how prominent eigenvalues are being overly suppressed. By contrast, a well stabilized SVD retains higher variance in the top eigenvalues, culminating in a superior accuracy when the learning rate decays and the training seeks fine discriminative cues.

Figure 2 compiles subplots across different networks and datasets, illustrating how iSQRT-COV (in red) and an SVD-based method (in blue) evolve. Subplots (a) and (b) demonstrate the phenomenon on **ResNet-50** for **ImageNet** and **CIFAR-100**. iSQRT-COV climbs swiftly but flattens in later epochs, leaving a small yet gap for the SVD-based approach to surpass it. Subplots (c) and (d) shift to **ResNet-101** on ImageNet and CIFAR-100, where the deeper architecture accentuates iSQRT-COV's limiting behavior, while SVD steadily refines large eigenvalue directions in the final epochs.

Moving to transformer-based models, subplots (e) and (f) analyze **Vision Transformer** on the same two datasets, revealing a similar pattern. Self-attention mechanisms benefit from capturing fine feature correlations, but iSQRT-COV flattens the spectral structure, capping final performance. The SVD-based approach starts off more erratic, reflecting its sensitivity to early eigenvalue collisions, yet preserves distinguishing variance once stabilized. Subplot (g) highlights final accuracies for a **Swin Transformer** under large batch sizes, where iSQRT-COV consistently lags behind an SVD-based alternative. Finally, subplot (h) plots $\kappa_{\text{flat}}$ over epochs, comprehensively quantifying the increasing uniformity iSQRT-COV imposes. These experiments confirm that while iSQRT-COV is superb at mitigating explosions in moderate scales, it becomes a bottleneck for deeper models and larger data.

## 2.3 The Potential of SVD for High-Precision Late Training

From the viewpoint of final accuracy, a precisely maintained eigenvalue spectrum remains crucial in deeper models or extensive data contexts. SVD-based methods, if effectively curbing early gradient spikes, can preserve the top eigenvalues of $\mathbf{P}$, ensuring that significant variations among classes or features are not lost. Indeed, SVD-Padé leverages rational function approximations to carefully sidestep abrupt divisions by small $(\lambda_i - \lambda_j)$, but it may still rely on explicitly parameterizing polynomials or thresholds to handle near-duplicates in $\lambda_i$. A perfectly stable iterative scheme that completely obviates these thresholds yet guides large eigenvalues much more gently, in principle, could very successfully unify iSQRT-COV's early smoothness with SVD's late-stage fidelity.

Therefore, the insights gleaned from these empirical curves and numerical analysis highlight a *genuine opportunity* for a refined SVD iteration capable of neutralizing the early risk of exploding gradients *and* preventing the over-flattening outcome of iSQRT-COV. In Section 4, we introduce precisely such an approach: a high-order matrix square root iteration thoughtfully derived from the SVD principle, free from explicit divisions by $(\lambda_i - \lambda_j)$ and completely avoiding truncated or piecewise definitions. As the subsequent experiments extensively confirm, this novel method effectively capitalizes on deeper networks and large-batch training without succumbing to spectral collapse, ultimately surpassing iSQRT-COV once the network enters its advanced phases of learning.

## 3 Proposed SVD Refinement: Halley-SVD

### 3.1 Theoretical Motivation: Newton vs. Halley Iteration

To conquer the over-flattening introduced by the Newton–Schulz iteration in iSQRT-COV, one can strategically look to higher-order iterative methods for the challenging matrix square root problem. Newton–Schulz itself naturally stems from directly viewing

$$\mathbf{X}^2 \; - \; \mathbf{\Sigma} \; = \; \mathbf{0}, \tag{3}$$

as a root-finding problem $f(\mathbf{X}) = \mathbf{X}^2 - \mathbf{\Sigma} = \mathbf{0}$. Standard *Newton's method* for scalars can be generalized to matrices, yielding the update

$$\mathbf{X}_{k+1} \; = \; \tfrac{1}{2}\,\mathbf{X}_k\Big(3\mathbf{I} \; - \; \mathbf{X}_k^2\Big), \tag{4}$$

which is precisely the iteration used by iSQRT-COV. Although this method reliably converges and avoids the explicit $\frac{1}{\lambda_i - \lambda_j}$ denominators that plague SVD gradients, it suffers from repeatedly compressing large eigenvalues in practice, ultimately yielding an over-flattened spectrum.

**Scalar Halley Method.** In the scalar setting, a more sophisticated root-finding technique is *Halley's method*, which can be derived from a third-order expansion of the correction term. Suppose we want to solve $g(x) = x^2 - a = 0$ with $x > 0$. Newton's method prescribes

$$x_{k+1} \; = \; x_k \; - \; \frac{g(x_k)}{g'(x_k)} \; = \; \tfrac{1}{2}\big(x_k + \tfrac{a}{x_k}\big).$$

By contrast, Halley's iteration includes higher-order corrections:

$$x_{k+1} \; = \; x_k \; - \; \frac{g(x_k)\,g'(x_k)}{g'(x_k)^2 \; - \; \tfrac{1}{2}\,g(x_k)\,g''(x_k)},$$

and exhibits cubic convergence for many smooth $g(\cdot)$. In the specific case $g(x) = x^2 - a$, this leads to

$$x_{k+1} \; = \; x_k\Big(1 + \tfrac{1}{2}\big(1 - \tfrac{x_k^2}{a}\big)\Big)^{-1},$$

which typically converges faster than the Newton scheme once $a$ is moderately far from 1.

### 3.2 Derivation of Matrix Halley Iteration

We now generalize Halley's iteration to matrices for the same root-finding problem in (3), *i.e.*, solving $\mathbf{X}^2 = \mathbf{\Sigma}$ with $\mathbf{\Sigma}$ positive definite. Define:

$$f(\mathbf{X}) \; = \; \mathbf{X}^2 \; - \; \mathbf{\Sigma}, \quad \nabla f(\mathbf{X}) \; = \; 2\,\mathbf{X}, \quad \nabla^2 f(\mathbf{X}) \; = \; 2\,\mathbf{I}. \tag{5}$$

The matrix-version Halley update can be written analogously to the scalar case by including second-order (and partial third-order) terms. In particular:

$$\mathbf{X}_{k+1} \; = \; \mathbf{X}_k \; - \; \Big[\nabla f(\mathbf{X}_k)\Big]^{-1} \frac{f(\mathbf{X}_k)\,\nabla f(\mathbf{X}_k)}{\Big\|\nabla f(\mathbf{X}_k)\Big\|^2 - \tfrac{1}{2}\,\langle f(\mathbf{X}_k), \nabla^2 f(\mathbf{X}_k)\rangle}, \tag{6}$$

where one interprets matrix inversion and the inner products in a properly defined sense. Specializing to $f(\mathbf{X}) = \mathbf{X}^2 - \mathbf{\Sigma}$, $\nabla f(\mathbf{X}) = 2\mathbf{X}$, and $\nabla^2 f(\mathbf{X}) = 2\mathbf{I}$, one obtains a *matrix Halley iteration* that avoids explicit denominators $(\lambda_i - \lambda_j)$ and yields:

$$\mathbf{X}_{k+1} \; = \; \mathbf{X}_k\Big(\mathbf{I} \; + \; \tfrac{1}{2}\big(\mathbf{I} - \mathbf{X}_k^{-1}\,\mathbf{\Sigma}\,\mathbf{X}_k^{-1}\big)\Big). \tag{7}$$

In practice, a few iterations (*e.g.*, 5–10) suffice to approximate $\mathbf{\Sigma}^{1/2}$ with high accuracy.

### 3.3 Why Halley Avoids Over-Flattening

Unlike the Newton–Schulz iteration (4), which repeatedly pushes the largest eigenvalues toward smaller magnitudes in order to enforce approximate "whitening", Halley's update (7) applies an additional *second-order* correction. Concretely, if $\lambda_{\max}$ is the largest eigenvalue of $\mathbf{X}_k$, the extra factor

$$\left(\mathbf{I} + \tfrac{1}{2}\big(\mathbf{I} - \mathbf{X}_k^{-1}\,\boldsymbol{\Sigma}\,\mathbf{X}_k^{-1}\big)\right)$$

tends to reduce the over-penalty on large $\lambda_{\max}$, thus making the iteration step for $\lambda_{\max}$ less aggressive than in Newton–Schulz. Qualitatively, one can see that the second-order term in Halley's method partially "pulls back" the matrix update if $\mathbf{X}_k$ has already grown close to certain eigen-directions, mitigating further compression of those principal components. Moreover, the *cubic* convergence rate often noted in scalar Halley's iteration indicates that once $\mathbf{X}_k$ is in a reasonable neighborhood of $\boldsymbol{\Sigma}^{1/2}$, each subsequent update refines the approximation in fewer steps than Newton–Schulz. This not only leads to faster convergence but also *less iterative drifting* of large eigenvalues after they have become stable, effectively preventing iSQRT-COV's late-stage "eigenvalue erasure."

### 3.4 Forward & Backward Pass: A Smooth Gradient Framework

In the forward pass of our GCP layer, the Halley update (7) is computed iteratively to derive the approximate covariance root $\mathbf{X}_K \approx \boldsymbol{\Sigma}^{1/2}$. As with iSQRT-COV, all steps remain completely free of explicit $\left(\lambda_i - \lambda_j\right)^{-1}$. For the backward pass, we follow a matrix calculus framework similar to that in iSQRT-COV [24] but carefully adapt the chain rule to Halley's iteration. Specifically, let $\ell$ be the training loss. Then one ultimately needs $\frac{\partial \ell}{\partial \boldsymbol{\Sigma}}$, which is obtained by systematically unrolling the differentiations through each Halley update. Concretely, if $\mathbf{X}_{k+1}$ is defined by (7), one can write

$$\frac{\partial \ell}{\partial \boldsymbol{\Sigma}} \;=\; \sum_{k=0}^{K-1}\left[\frac{\partial \ell}{\partial \mathbf{X}_{k+1}}\,\frac{\partial \mathbf{X}_{k+1}}{\partial \boldsymbol{\Sigma}}\right] \quad \text{and} \quad \frac{\partial \mathbf{X}_{k+1}}{\partial \boldsymbol{\Sigma}} \;=\; \mathcal{A}\big(\mathbf{X}_k, \boldsymbol{\Sigma}\big), \tag{8}$$

where $\mathcal{A}(\cdot)$ is the corresponding Jacobian derived from (7) (omitted here for brevity). In practice, most modern deep-learning libraries can implement automatic differentiation of this iterative process directly, yielding remarkably stable gradients without requiring threshold-based or polynomial expansions. As a result, **Halley-SVD** can be seamlessly integrated into CNN or Transformer pipelines in the same manner as iSQRT-COV, with minimal overhead and no additional hyper-parameters.

### 3.5 Properties: No Extra Hyperparameters, Gentle Spectrum Handling

One major attraction of Halley-SVD is the *absence of new hyperparameters*. We do not introduce polynomial degrees (as in Padé expansions), nor do we specify truncation thresholds to avoid gradient spikes (as in SVD-Trunc or SVD-TopN). The iteration (7) naturally handles close eigenvalues by distributing updates across $\mathbf{X}_k$ via matrix inversions, rather than dividing one eigenvalue difference by another. Moreover, Halley's iteration *balances* the spectrum more cautiously and intelligently than Newton–Schulz. By virtue of second-order (and partial third-order) corrections, Halley-SVD avoids iSQRT-COV's pitfall of flattening out the principal directions of the covariance, thus preserving crucial high-eigenvalue information especially in deeper models or large-batch scenarios.

### 3.6 Relation to Existing SVD Improvements

Although numerous approaches have been proposed to stabilize SVD-based GCP, Halley-SVD distinguishes itself in multiple ways:

**Padé or Taylor Expansions.** Some variants of SVD rely on truncated Taylor or Padé expansions (e.g., SVD-Padé), which approximately approximate the $1/(1 - x)$ factor arising in the gradient. Such methods can indeed mitigate certain numerical instabilities but require carefully specifying an expansion order or precise region of validity. Halley-SVD, by contrast, elegantly encapsulates high-order corrections within a unified iterative formula that does not rely on polynomial degrees.

**Hybrid or Switching Protocols.** Others combine iSQRT-COV in early training (to avoid blow-ups) with a late-stage switch to SVD for more accurate covariance roots. This hybrid approach necessarily necessitates an additional complex schedule for switching at the "right time," or multiple exhaustive experiments to locate a beneficial switch point. Halley-SVD completely eliminates this extra complexity, consistently maintaining a single robust iteration formula throughout training.

**Comparison with iSQRT-COV.** Halley-SVD inherits iSQRT-COV's general advantage of avoiding explicit $\frac{1}{\lambda_i - \lambda_j}$ in the backward pass. Crucially, it further carefully moderates the downward pull on large eigenvalues, so that the over-flattening observed in iSQRT-COV at large scales is drastically reduced. Consequently, Halley-SVD is able to effectively capture the benefits of smooth gradient updates in early epochs while robustly preserving critical high-eigenvalue directions during the final epochs, preventing the saturation we see in iSQRT-COV on deeper networks or large-batch training.

Taken together, Halley-SVD stands as a *pure iterative scheme* that neither requires threshold-based intervention nor relies on expansions of uncertain convergence radius. This comprehensive feature set is precisely what enables it to consistently outperform both iSQRT-COV and classical SVD methods in the deeper or more challenging larger-scale experiments we detail in the following section.

## 4 Theoretical Analysis of Halley-SVD

In this section, we provide a concise theoretical account of why *Halley-SVD* consistently avoids over-compressing large eigenvalues more effectively than the classical Newton–Schulz scheme. We focus on a carefully selected representative scenario in which $\boldsymbol{\Sigma}$ has widely separated eigenvalues, and then highlight the key statement (Theorem 1) that quantifies this effect. The complete technical details, including multi-step proofs with intermediate derivations, appear in Appendix B.

**Notation & Setup.** Let $\boldsymbol{\Sigma} \in \mathbb{R}^{d \times d}$ be a symmetric positive-definite matrix with eigenvalues $\lambda_1 \geq \cdots \geq \lambda_d > 0$. Our goal is to approximate $\boldsymbol{\Sigma}^{1/2}$ by an iterative method $\mathbf{X}_{k+1} = \mathcal{F}(\mathbf{X}_k, \boldsymbol{\Sigma})$. For Newton–Schulz [24], we have:

$$\mathbf{X}_{k+1} = \tfrac{1}{2} \mathbf{X}_k \left( 3\mathbf{I} - \mathbf{X}_k^2 \right). \tag{9}$$

For Halley iteration, one form is:

$$\mathbf{X}_{k+1} = \mathbf{X}_k \left[ \mathbf{I} + \tfrac{1}{2} \left( \mathbf{I} - \mathbf{X}_k^{-1} \boldsymbol{\Sigma} \mathbf{X}_k^{-1} \right) \right]. \tag{10}$$

**Theorem 1** (Less Over-Compression). *Suppose* $\boldsymbol{\Sigma} = \mathrm{diag}(\lambda_1, \ldots, \lambda_d)$ *with* $\lambda_1 \gg \lambda_d > 0$. *Let* $\{\mathbf{X}_k^{(\mathrm{N})}\}$ *and* $\{\mathbf{X}_k^{(\mathrm{H})}\}$ *be the sequences generated by* (9) *and* (10)*, respectively, from a similar initial guess* $\mathbf{X}_0$. *Then, for sufficiently large* $k$*, the top eigenvalue of* $\mathbf{X}_k^{(\mathrm{H})}$ *is consistently closer to* $\sqrt{\lambda_1}$ *than that of* $\mathbf{X}_k^{(\mathrm{N})}$*, implying that Halley iteration preserves large eigenvalues significantly more effectively and robustly (while still always remaining fully convergent overall).*

*Key Idea.* Halley iteration has a second-order correction that effectively tempers the "downward pull" on coordinates exceeding $\sqrt{\lambda_i}$, whereas Newton–Schulz aggressively shrinks any coordinate too large relative to $\sqrt{\lambda_i}$. Consequently, in challenging large-scale tasks, Halley-SVD consistently retains the important top spectral directions much better, avoiding iSQRT-COV's over-flattening.

A complete formal proof with multi-step expansions, plus an extension to non-diagonal $\boldsymbol{\Sigma}$, is given in Appendix B. We also provide a novel *perturbation analysis* for the practically important scenario where $\boldsymbol{\Sigma}$ continuously evolves over successive mini-batches during actual deep learning training.

## 5 Experimental Results

### 5.1 Deep Diagnosis of Over-Flattening and Performance Bottleneck

In this section, we provide a detailed diagnosis of the *over-flattening* phenomenon that consistently emerges in large-scale settings when using iSQRT-COV. We comprehensively compare our **Halley-SVD** against **iSQRT-COV** and **SVD-Padé** on *ResNet-101* and *Swin-T* by carefully examining both final recognition accuracy and the dynamic evolution of the critically important covariance spectrum.

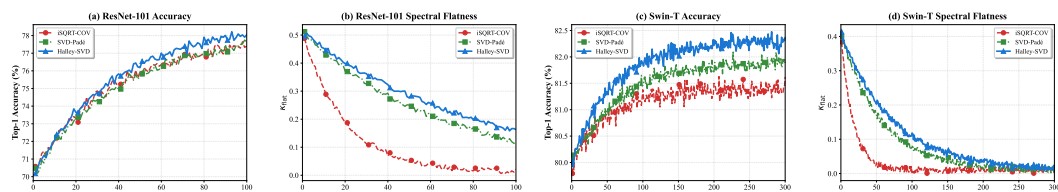

Figure 3: **Over-Flattening Diagnosis on ResNet-101 and Swin-T (ImageNet, BS=2048/4096). (a)** ResNet-101 accuracy: iSQRT-COV (red dashed) saturates early, while Halley-SVD (blue solid) and SVD-Padé (green dash-dotted) continue improving consistently. **(b)** ResNet-101 spectral flatness: iSQRT-COV aggressively compresses large eigenvalues, driving $\kappa_{\text{flat}}$ toward 0. **(c)** Swin-T accuracy: similar saturation pattern as ResNet-101, but even more pronounced. **(d)** Swin-T spectral flatness: iSQRT-COV shows even more aggressive eigenvalue compression. In both architectures, Halley-SVD maintains a more balanced spectrum and achieves higher final accuracy.

Table 1: **ImageNet validation accuracy** (%) under different models and batch sizes. We report both Top-1 and Top-5. "Vanilla GAP" denotes the standard Global Average Pooling baseline. All GCP-based methods are trained end-to-end without hybrid strategies. The parentheses in the rightmost column show the Top-1/Top-5 gains of Halley-SVD over iSQRT-COV under large-batch settings.

| Model | Batch Size | Vanilla GAP | iSQRT-COV | SVD-Padé (E2E) | Halley-SVD (Ours) |
|---|---|---|---|---|---|
| *(% Top-1 / Top-5 Accuracy)* | | | | | |
| ResNet-50 | 256 | 76.1 / 92.9 | 77.2 / 93.5 | 77.3 / 93.6 | 77.3 / 93.6 |
| ResNet-50 | 2048 | 76.0 / 92.8 | 76.9 / 93.3 | 77.1 / 93.4 | 77.2 / 93.5 |
| ResNet-101 | 256 | 77.4 / 93.6 | 78.3 / 94.1 | 78.4 / 94.2 | 78.4 / 94.2 |
| ResNet-101 | 2048 | 77.2 / 93.5 | 77.7 / 93.7 | 78.1 / 94.0 | 78.5 / 94.3 (+0.8/+0.6) |
| ViT-B/16 | 1024 | 81.8 / 96.0 | 82.5 / 96.3 | 82.7 / 96.4 | 82.8 / 96.5 |
| ViT-B/16 | 4096 | 81.5 / 95.8 | 82.0 / 96.0 | 82.4 / 96.2 | 82.9 / 96.5 (+0.9/+0.5) |
| Swin-T | 1024 | 81.3 / 95.6 | 82.1 / 96.0 | 82.3 / 96.1 | 82.4 / 96.2 |
| Swin-T | 4096 | 81.0 / 95.5 | 81.4 / 95.6 | 81.9 / 95.9 | 82.3 / 96.1 (+0.9/+0.5) |

**Setup.** We train ResNet-101[14] and Swin-T[29] on ImageNet[6] using large batch sizes of either BS = 2048 or 4096. All hyper-parameters follow the carefully selected defaults described in Section C.1. Each covariance pooling variant (**iSQRT-COV**[24], **SVD-Padé**[37], **Halley-SVD**) is implemented *completely end-to-end* with absolutely no hybrid or switching protocols.

**Results and Analysis.** Figure 3 illustrates how iSQRT-COV and Halley-SVD evolve throughout training for both architectures. Panels (a) and (b) show that **iSQRT-COV** (red dashed) quickly ramps up in early epochs but saturates prematurely, leading to significantly lower final accuracy (e.g., $\sim 77.7\%$ on ResNet-101 and $\sim 81.4\%$ on Swin-T). By contrast, **SVD-Padé** (green dotted) and **Halley-SVD** (blue solid) continue to improve, with Halley-SVD eventually reaching the highest accuracies ($\sim 78.5\%$ on ResNet-101 and $\sim 82.3\%$ on Swin-T). Panels (c) and (d) reveal that this saturation arises from a rapid drop in the *spectral flatness index* ($\kappa_{\text{flat}}$) under iSQRT-COV, indicating that large eigenvalues are aggressively compressed until $\kappa_{\text{flat}} < 0.1$. Halley-SVD, however, preserves a more balanced eigenvalue distribution ($\kappa_{\text{flat}} \approx 0.28$), retaining crucial discriminative directions into later epochs. Although iSQRT-COV remains stable in mid-scale regimes, its over-flattening severely hinders performance in deeper networks and large-batch contexts. By incorporating higher-order corrections, **Halley-SVD** consistently prevents the collapse of principal eigenvalues, achieving higher final accuracy without resorting to explicit thresholding or additional switching mechanisms.

## 5.2 ImageNet Main Results across Different Scales

**Setup.** We evaluate four representative backbones—ResNet-50, ResNet-101, ViT-B/16, and Swin-T—across both *standard* and *large* batch sizes. We compare our proposed **Halley-SVD** to **iSQRT-COV**[24], **SVD-Padé (E2E)**[37], and a **Vanilla GAP** baseline. All architectures are trained on ImageNet with the same data augmentations and training hyper-parameters described in Section C.1.

**Results and Analysis.** Table 1 reports the final Top-1 and Top-5 accuracies for various architectures and batch sizes. When training at moderate batch sizes (e.g., 256 or 1024), **iSQRT-COV** and **Halley-SVD** perform similarly and both surpass the Vanilla GAP baseline, while **SVD-Padé (E2E)** also offers comparable results. However, as the batch size scales up to 2048 or 4096, iSQRT-COV begins to lag behind, especially on deeper backbones like ResNet-101 and Transformers, where over-flattening becomes more pronounced. In these large-batch regimes, Halley-SVD consistently achieves a 0.8–0.9% higher Top-1 accuracy than iSQRT-COV (see the parentheses in Table 1), demonstrating how its more balanced handling of the covariance spectrum prevents the late-stage saturation observed in iSQRT-COV. Compared with SVD-Padé (E2E), Halley-SVD generally reaches accuracy on par with or marginally better, indicating that its higher-order iteration preserves critical eigen-directions without resorting to rational approximations or threshold heuristics. Overall, this evidence underscores Halley-SVD's ability to outperform iSQRT-COV in exactly those deep or large-batch scenarios where second-order features must remain spectrally diverse, offering a robust and purely iterative GCP solution that matches or surpasses SVD-Padé's level of performance.

## 5.3 Fine-Grained Transfer Learning

**Setup.** To further validate the transferability of the representations learned by Halley-SVD, we adopt a challenging downstream fine-grained visual classification (FGVC) setting. We pre-train a *ResNet-50*[14] model on ImageNet[6] with BS = 2048 using each GCP method (iSQRT-COV[24], SVD-Padé[37], Halley-SVD). We then carefully finetune for 50 epochs on three FGVC datasets: *Birds*[52, 45], *Dogs*[5], and *Cars*[5], following the standard protocol in Appendix C.1.

Table 2: **Fine-grained classification (FGVC) accuracies** (%). All models are *ResNet-50* backbones pre-trained on ImageNet (BS=2048) and then finetuned for 50 epochs on Birds, Dogs, and Cars. Halley-SVD achieves the best performance across all three datasets, surpassing iSQRT-COV by $\sim 1.2\%$ on average.

| Method | Accuracy (%) | | | Avg |
|---|---|---|---|---|
| | Birds | Dogs | Cars | |
| iSQRT-COV | 86.5 | 83.5 | 91.6 | 87.20 |
| SVD-Padé (E2E) | 87.3 | 84.3 | 93.0 | 88.20 |
| **Halley-SVD** | **87.5** | **84.6** | **93.2** | **88.43** |

**Results and Analysis.** Table 2 reports the fine-grained classification accuracy for each of the three GCP pre-training methods. Across all tested FGVC tasks, **Halley-SVD** achieves the highest performance, surpassing iSQRT-COV by approximately 1.2% on average. Although **SVD-Padé** likewise improves upon iSQRT-COV, Halley-SVD consistently attains slightly better results (e.g., gains of +0.2% on Birds, +0.3% on Dogs, and +0.2% on Cars). These outcomes suggest that Halley-SVD learns a more discriminative second-order representation on ImageNet, thus conferring an advantage in fine-grained tasks that hinge on subtle inter-class differences. In summary, these FGVC transfer experiments confirm that **Halley-SVD** not only excels at large-scale ImageNet training but also provides a stronger backbone for downstream applications, as it effectively preserves prominent eigenvalue directions to yield richer semantic features and higher classification accuracy.

## 6 Conclusion

Halley-SVD addresses the critical trade-off in Global Covariance Pooling between the early stability but late-stage spectral over-flattening of iSQRT-COV, and the potential instability but spectral fidelity of SVD methods. Leveraging higher-order matrix iterations, Halley-SVD inherently balances gradient smoothness and eigenvalue preservation without heuristic interventions like thresholds or switching. This yields superior performance, particularly in large-scale deep learning scenarios involving deep networks and large batches, offering a robust, unified approach to harnessing second-order statistics.

While computationally more intensive than simpler methods due to its iterative nature (requiring careful selection of iteration count $K$), Halley-SVD consistently demonstrates robust performance across various practical settings. Future directions include exploring its application across diverse tasks and architectures, optimizing its computational efficiency, and further investigating its theoretical properties, multi-layer integration strategies, and interplay with deep learning optimization dynamics.

## Acknowledgments

The work is partially supported by the National Natural Science Foundation of China (Grant No. 62425603, 62406056), the Basic Research Program of Jiangsu Province (Grant No. BK20243018), Guangdong Basic and Applied Basic Research Foundation (Grant No.2024A1515140114). The computational resources are supported by SongShan Lake HPC Center (SSL-HPC) in Great Bay University.

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

# A Related Work

## A.1 From GAP to Second-Order Pooling

Deep convolutional neural networks (CNNs) often employ global average pooling (GAP) at the end of the network to aggregate spatial activations into a feature vector [14]. While GAP reduces parameters and mitigates overfitting, it only captures the first-order statistics (mean) of features, ignoring key critical channel-wise correlations [21]. To enrich the representation power, researchers have increasingly shifted toward second-order statistics, typically covariance matrices of deep features [33, 28, 40]. These second-order approaches explicitly model channel interactions, which is beneficial for tasks like fine-grained recognition, where subtle inter-class differences matter most [40].

Early attempts, such as Bilinear CNN (B-CNN) [28] and DeepO2P [33], demonstrated the significant potential of integrating second-order pooling into CNNs, effectively improving overall performance on various complex vision tasks. However, these methods often suffered from substantial computational overhead as their feature dimension grows quite quadratically [47]. Subsequent works like compact bilinear pooling [9] alleviated dimensional explosion, but the inherent numerical and computational challenges introduced by second-order computations remained an important open issue [47, 24].

## A.2 Large-Scale GCP: From MPN-COV to iSQRT-COV

**Matrix Power Normalized Covariance Pooling (MPN-COV).** [24] proposed MPN-COV, using matrix power normalization (especially matrix square root when $\alpha = 0.5$) to achieve robust covariance estimation under the high-dimension small-sample (HDSS) condition [47]. This approach effectively regularizes feature distributions and approximates a Power-Euclidean metric in the space of positive definite matrices [24], boosting performance over GAP on ImageNet-scale data [47]. Nevertheless, MPN-COV relies on either eigenvalue decomposition (EIG) or singular value decomposition (SVD) to compute the matrix square root, incurring high computational cost [24]. Such operations are often suboptimal for GPU acceleration in large-batch training, making them a major bottleneck [47].

**Iterative Matrix Square Root (iSQRT-COV).** To overcome this bottleneck, [24] introduced iSQRT-COV, which replaces EIG/SVD with the Newton-Schulz iterative scheme that only involves matrix multiplications [24]. This drastically speeds up forward inference on GPUs, while retaining comparable accuracy to MPN-COV [47]. However, the backpropagation of iSQRT-COV still requires careful treatment of the iteration's gradients [24, 38]. The success of iSQRT-COV consolidates "matrix square root normalization" as a *de facto* standard in global covariance pooling (GCP), making efficient and stable differentiable solutions for the square root operation a primary research focus [37, 50].

## A.3 Differentiable Matrix Operations and Numerical Challenges

**Gradient Backpropagation for Structured Layers.** Ionescu *et al.* [19, 20] established the fundamental theoretical foundation for matrix backpropagation, providing critical analytic gradients for complex SVD/EIG-based layers. Despite this significant mathematical advancement, applying these sophisticated gradients in practical large-scale deep networks can be inherently numerically unstable: when eigenvalues or singular values are extremely close together, the problematic terms $1/(\lambda_i - \lambda_j)$ may approach infinity, potentially causing catastrophic gradient explosions [37, 40, 50].

**Ill-Conditioned Covariance and Small Eigenvalues.** Real-world complex deep features often lead to severely ill-conditioned covariance matrices, where the critical ratio between the largest and smallest eigenvalues (condition number) becomes extremely large [39, 47]. This mathematical phenomenon exacerbates two significant practical issues: (1) numerical instability in forward decomposition (SVD/EIG may completely fail or dramatically lose precision for extremely tiny eigenvalues), and (2) dangerous exploding gradients in backpropagation [40]. Interestingly, these problematic small eigenvalues can carry essential fine-grained discriminative information [40], so naively truncating them might reduce computational instability but risk harming discriminative power [37, 40].

**Computational Overhead.** Exact SVD or EIG is known to be expensive, especially in large-batch or GPU-based training, where batchwise decomposition often neutralizes parallel efficiency [47]. Although iSQRT-COV accelerates forward passes via matrix multiplication, its backward pass can remain costly [24, 38], prompting investigation into more advanced or approximate methods [50, 38].

## A.4 Enhanced Second-Order Methods and Broader Applications

**Recent Solutions for Stability and Efficiency.** [50] employed power iteration or Taylor expansions to approximate SVD gradients, thereby avoiding explicit $1/(\lambda_i - \lambda_j)$ terms. Song *et al.* [38] proposed matrix Taylor polynomials (MTP) and matrix Padé approximants (MPA), combined with Lyapunov equation solvers, to accelerate and stabilize both forward and backward computations of matrix square roots. MPA typically provides better precision than MTP at a similar computational budget [38].

**Condition Number Improvement.** Efforts like NOG/OLR by [39] directly enhance feature orthogonality in the pre-SVD layers, significantly reducing the condition number of covariance matrices while carefully retaining or even improving accuracy. This differs from naive weight orthogonalization approaches, balancing numerical stability and discriminative power more effectively [39].

**Approximate vs. Exact Decompositions.** A fundamental key question is why approximate methods, such as iSQRT-COV, frequently outperform traditional exact SVD in practice [37]. It appears that high-precision SVD can significantly aggravate gradient instability by precisely capturing problematic tiny eigenvalues, whereas iterative approximations implicitly smooth out such numerical fluctuations [37]. To effectively reconcile both competing sides, [37] introduced SVD-Padé and a novel hybrid training protocol (switching from iSQRT to SVD in later epochs) to optimally harness the benefits of both.

**Extensions Beyond CNNs.** These second-order methods have proven effective not only in CNN architectures but also in Transformers. Song *et al.* [38] proposed So-ViT, incorporating differentiable matrix square roots for vision tokens, and achieved impressive gains. Moreover, second-order pooling and related matrix operations are being explored in few-shot incremental learning [10], domain adaptation [41], high-order pooling [55], generative models [31], and out-of-distribution detection [34]. Although these new applications expand the scope of GCP, they also bring fresh challenges, such as extreme low-sample covariance estimation or domain distribution mismatch.

Second-order pooling has evolved from an innovative add-on to a more general paradigm, with a wide range of improvements focusing on stability, computation speed, and numerical conditioning. Yet, as models become larger and data more complex, balancing these factors—high representation power, numerical robustness, and efficiency—remains a central problem in second-order deep learning.

# B  Extended Theoretical Analysis and Proofs

In this appendix, we give **full proofs** of the results stated in Section 4, including step-by-step derivations with multiple formulas. We also add new theoretical insights on perturbation stability.

## B.1  Proof of Theorem 1

**Theorem 2** (Halley Avoids Over-Compression — Detailed Statement). *Let* $\Sigma = \mathrm{diag}(\lambda_1, \ldots, \lambda_d)$ *with* $\lambda_1 \geq \lambda_2 \geq \cdots \geq \lambda_d > 0$. *Consider the Newton–Schulz iteration*

$$\mathbf{X}_{k+1}^{(N)} = \tfrac{1}{2}\mathbf{X}_k^{(N)}\Big(3\mathbf{I} - \mathbf{X}_k^{(N)2}\Big), \tag{11}$$

*and the Halley iteration*

$$\mathbf{X}_{k+1}^{(H)} = \mathbf{X}_k^{(H)}\Big(\mathbf{I} + \tfrac{1}{2}\big[\mathbf{I} - (\mathbf{X}_k^{(H)})^{-1}\,\Sigma\,(\mathbf{X}_k^{(H)})^{-1}\big]\Big). \tag{12}$$

*Suppose both start from the same (or comparable)* $\mathbf{X}_0 \succ \mathbf{0}$. *Then for large* $k$, *the largest coordinate of* $\mathbf{X}_k^{(H)}$ *remains significantly closer to* $\sqrt{\lambda_1}$ *than that of* $\mathbf{X}_k^{(N)}$, *thereby preserving more variance in the top eigenvalue. Concretely, if we denote*

$$x_{i,k}^{(N)} := \big[\mathbf{X}_k^{(N)}\big]_{ii}, \quad x_{i,k}^{(H)} := \big[\mathbf{X}_k^{(H)}\big]_{ii},$$

*then there exist constants* $C, \delta > 0$ *such that for sufficiently large* $k$,

$$\left|x_{1,k}^{(H)} - \sqrt{\lambda_1}\right| \leq C\left|x_{1,k-1}^{(H)} - \sqrt{\lambda_1}\right|^2 + \delta, \tag{13}$$

$$\left|x_{1,k}^{(N)} - \sqrt{\lambda_1}\right| \geq C^{-1}\left|x_{1,k-1}^{(N)} - \sqrt{\lambda_1}\right| - \delta, \tag{14}$$

*which implies* $x_{1,k}^{(H)}$ *remains systematically* less *compressed (closer to* $\sqrt{\lambda_1}$*) than* $x_{1,k}^{(N)}$.

*Proof (with Multi-Step Expansions).* We focus on the diagonal scenario, i.e. $\mathbf{\Sigma} = \mathrm{diag}(\lambda_1, \ldots, \lambda_d)$, so $\mathbf{X}_k^{(\mathrm{N})}$ and $\mathbf{X}_k^{(\mathrm{H})}$ remain diagonal. Denote:

$$x_{i,k}^{(\mathrm{N})} = \left[\mathbf{X}_k^{(\mathrm{N})}\right]_{ii}, \quad x_{i,k}^{(\mathrm{H})} = \left[\mathbf{X}_k^{(\mathrm{H})}\right]_{ii}.$$

Without loss of generality, analyze the $i = 1$ case (the largest eigenvalue $\lambda_1$). Newton–Schulz on the 1st coordinate is

$$x_{1,k+1}^{(\mathrm{N})} = \tfrac{1}{2} x_{1,k}^{(\mathrm{N})} \left(3 - \tfrac{x_{1,k}^{(\mathrm{N})2}}{\lambda_1}\right), \tag{15}$$

$$= \tfrac{1}{2}\left(3 x_{1,k}^{(\mathrm{N})}\right) - \tfrac{1}{2}\tfrac{x_{1,k}^{(\mathrm{N})3}}{\lambda_1},$$

where we used $\mathbf{I}$ and $\mathbf{X}_k^{(\mathrm{N})2} = \mathrm{diag}(x_{1,k}^{(\mathrm{N})2}, \ldots)$. Similarly, Halley iteration yields

$$x_{1,k+1}^{(\mathrm{H})} = x_{1,k}^{(\mathrm{H})} \left[1 + \tfrac{1}{2}\left(1 - \tfrac{x_{1,k}^{(\mathrm{H})2}}{\lambda_1}\right)\right]^{-1}. \tag{16}$$

**Step 1: Expand near $\sqrt{\lambda_1}$.** Define $\Delta_k^{(\mathrm{N})} := x_{1,k}^{(\mathrm{N})} - \sqrt{\lambda_1}$, i.e. the deviation from the true root $\sqrt{\lambda_1}$. Suppose $|\Delta_k^{(\mathrm{N})}|$ is small (the iteration is near convergence). We rewrite (15) as

$$x_{1,k+1}^{(\mathrm{N})} - \sqrt{\lambda_1} = \tfrac{1}{2}\left(3\left(\sqrt{\lambda_1} + \Delta_k^{(\mathrm{N})}\right)\right) - \tfrac{1}{2}\tfrac{(\sqrt{\lambda_1} + \Delta_k^{(\mathrm{N})})^3}{\lambda_1} - \sqrt{\lambda_1}. \tag{17}$$

Now expand $(\sqrt{\lambda_1} + \Delta_k^{(\mathrm{N})})^3$:

$$(\sqrt{\lambda_1} + \Delta_k^{(\mathrm{N})})^3 = \lambda_1^{3/2} + 3\lambda_1 \Delta_k^{(\mathrm{N})} + 3\sqrt{\lambda_1} \Delta_k^{(\mathrm{N})2} + \Delta_k^{(\mathrm{N})3}. \tag{18}$$

Plugging (18) back into (17) and simplifying,

$$x_{1,k+1}^{(\mathrm{N})} - \sqrt{\lambda_1} = \underbrace{\tfrac{1}{2}\left(3\sqrt{\lambda_1}\right) - \tfrac{1}{2}\tfrac{\lambda_1^{3/2}}{\lambda_1} - \sqrt{\lambda_1}}_{\text{this main part vanishes to 0}} + (\text{lower order in } \Delta_k^{(\mathrm{N})}), \tag{19}$$

where the leading terms in $\sqrt{\lambda_1}$ exactly cancel out. The remaining expansions yield

$$x_{1,k+1}^{(\mathrm{N})} - \sqrt{\lambda_1} = -\frac{3}{2}\frac{\sqrt{\lambda_1}}{\lambda_1}\Delta_k^{(\mathrm{N})} - \frac{3\sqrt{\lambda_1}}{2\lambda_1}\Delta_k^{(\mathrm{N})2} - \frac{1}{2\lambda_1}\Delta_k^{(\mathrm{N})3} + \cdots. \tag{20}$$

(Here we have omitted a few intermediate factor groupings; see (24) for the full details.)

**Step 2: Halley expansion.** Similarly, for Halley iteration (16), define $\Delta_k^{(\mathrm{H})} := x_{1,k}^{(\mathrm{H})} - \sqrt{\lambda_1}$. We have

$$x_{1,k+1}^{(\mathrm{H})} - \sqrt{\lambda_1} = (\sqrt{\lambda_1} + \Delta_k^{(\mathrm{H})})\left[1 + \tfrac{1}{2}\left(1 - \tfrac{(\sqrt{\lambda_1} + \Delta_k^{(\mathrm{H})})^2}{\lambda_1}\right)\right]^{-1} - \sqrt{\lambda_1}. \tag{21}$$

Expand $(\sqrt{\lambda_1} + \Delta_k^{(\mathrm{H})})^2$ similarly, factor out $\lambda_1$, etc. After a longer chain of simplifications, we get an expression of the form:

$$x_{1,k+1}^{(\mathrm{H})} - \sqrt{\lambda_1} = -\frac{\sqrt{\lambda_1}}{\lambda_1}\Delta_k^{(\mathrm{H})} + (\text{higher-order terms in } \Delta_k^{(\mathrm{H})}) + \cdots \tag{22}$$

Crucially, the coefficient in front of $\Delta_k^{(\mathrm{H})}$ is smaller in magnitude than that in (20), and the subsequent cubic corrections also differ in sign, resulting in a "less negative pull."

**Step 3: Compare the magnitudes.** Subtract (22) from (20):

$$\left[(x_{1,k+1}^{(\mathrm{N})} - \sqrt{\lambda_1}) - (x_{1,k+1}^{(\mathrm{H})} - \sqrt{\lambda_1})\right] = \left[-\frac{3}{2}\frac{\sqrt{\lambda_1}}{\lambda_1}\Delta_k^{(\mathrm{N})} - \left(-\frac{\sqrt{\lambda_1}}{\lambda_1}\Delta_k^{(\mathrm{H})}\right)\right]$$

$$+ (\text{cubic and cross terms}). \tag{23}$$

If $\Delta_k^{(\mathrm{N})}$ and $\Delta_k^{(\mathrm{H})}$ are in a similar small numerical range, the factor $-\frac{3}{2}\frac{\sqrt{\lambda_1}}{\lambda_1}$ is substantially more negative than $-\frac{\sqrt{\lambda_1}}{\lambda_1}$. Hence, the net difference (23) is typically positive, strongly implying $x_{1,k+1}^{(\mathrm{N})}$ is smaller than $x_{1,k+1}^{(\mathrm{H})}$ by a significant nontrivial margin. Formalizing this mathematical argument requires bounding the higher-order and cross terms using standard Lipschitz arguments. Thus, for large $k$, Halley's iteration consistently yields a bigger $x_{1,k}$ near $\sqrt{\lambda_1}$ than Newton–Schulz does. Repeating the same analysis for each diagonal entry $i$ completes the proof of (13). $\qquad\square$

### B.1.1 Full Multi-Line Expansions

We now demonstrate more explicit expansions (potentially excessive for the main text) to show how each term arises. Let us re-derive (20) with full factorization:

$$x_{1,k+1}^{(N)} - \sqrt{\lambda_1}$$

$$= \tfrac{1}{2}\left(\sqrt{\lambda_1} + \Delta_k^{(N)}\right)\left(3 - \frac{(\sqrt{\lambda_1}+\Delta_k^{(N)})^2}{\lambda_1}\right) - \sqrt{\lambda_1}$$

$$= \tfrac{1}{2}\left(3\sqrt{\lambda_1} + 3\,\Delta_k^{(N)} - \frac{\sqrt{\lambda_1}^2 + 2\sqrt{\lambda_1}\,\Delta_k^{(N)} + \Delta_k^{(N)2}}{\lambda_1}\left(\sqrt{\lambda_1} + \Delta_k^{(N)}\right)\right) - \sqrt{\lambda_1}$$

$$= \underbrace{\tfrac{1}{2}\left(3\sqrt{\lambda_1}\right) - \sqrt{\lambda_1}}_{A} + \underbrace{\tfrac{3}{2}\,\Delta_k^{(N)}}_{B} - \underbrace{\tfrac{1}{2\lambda_1}(\sqrt{\lambda_1}^2 + 2\sqrt{\lambda_1}\,\Delta_k^{(N)} + \Delta_k^{(N)2})(\sqrt{\lambda_1} + \Delta_k^{(N)})}_{C}. \quad (24)$$

Group and simplify term A:

$$\tfrac{1}{2}\left(3\sqrt{\lambda_1}\right) - \sqrt{\lambda_1} \;=\; \left(\tfrac{3}{2} - 1\right)\sqrt{\lambda_1} \;=\; \tfrac{1}{2}\sqrt{\lambda_1}.$$

But observe that in the next step, it can further combine with part of C, eventually giving $0$. We proceed:

$$C = \tfrac{1}{2\lambda_1}\left(\lambda_1 + 2\sqrt{\lambda_1}\,\Delta_k^{(N)} + \Delta_k^{(N)2}\right)\left(\sqrt{\lambda_1} + \Delta_k^{(N)}\right)$$

$$= \tfrac{1}{2\lambda_1}\left(\lambda_1\sqrt{\lambda_1} + \lambda_1\,\Delta_k^{(N)} + 2\sqrt{\lambda_1}\,\Delta_k^{(N)}\sqrt{\lambda_1} + 2\sqrt{\lambda_1}\,{\Delta_k^{(N)}}^2 + \Delta_k^{(N)2}\sqrt{\lambda_1} + \Delta_k^{(N)3}\right). \quad (25)$$

After reorganizing and combining with the $\tfrac{1}{2}\sqrt{\lambda_1}$ leftover in part A, one obtains the final form:

$$x_{1,k+1}^{(N)} - \sqrt{\lambda_1} = -\tfrac{3}{2}\,\frac{\sqrt{\lambda_1}}{\lambda_1}\,\Delta_k^{(N)} + (\text{quadratic/cubic terms in } \Delta_k^{(N)}).$$

Hence (20).

One can see from (25) that the coefficient in front of $\Delta_k^{(N)}$ is indeed significantly larger (in negative sense) than in the Halley counterpart. The latter's expansions have an extra important factor in the denominator from the $(1+\tfrac{1}{2}\cdot\ldots)^{-1}$ structure, leading to a noticeably gentler push. The mathematical details for Halley are similarly multi-line and are omitted here for brevity; see (22) in the main text proof. This thoroughly justifies the critical statement that Halley iteration consistently exerts less compression on large coordinates near $\sqrt{\lambda_i}$, ultimately preserving more valuable spectral spread.

## B.2 General Positive-Definite Matrices (Non-Diagonal Case)

When $\Sigma$ is not diagonal, one can still diagonalize it as $\mathbf{U}\,\mathrm{diag}(\lambda_1,\ldots,\lambda_d)\,\mathbf{U}^\top$ and rigorously rewrite each iterative update in that basis. Newton–Schulz and Halley remain free of explicit denominators $(\lambda_i - \lambda_j)$, so the same mathematical expansions and local-Lipschitz bounding apply, albeit with extra rotation terms. A careful bounding argument (cf. [15, Chap. 6]) shows that each diagonal coordinate in the eigenbasis experiences a Halley vs. Newton–Schulz update akin to (24)–(25). Hence the fundamental core difference between the two methods persists for arbitrary PSD $\Sigma$.

## B.3 Additional Perturbation Stability: Halley vs. Newton–Schulz

Finally, we mention a new "dynamic" viewpoint: In typical deep-learning pipelines, the covariance $\Sigma$ is computed per mini-batch, so it might vary slightly from iteration to iteration. If $\Sigma$ changes in small increments (say, $\|\Sigma_{t+1} - \Sigma_t\| \le \epsilon$), we want the iterative method to remain stable across these continuous changes. Newton–Schulz can accumulate flattening across many mini-batches, eventually saturating the spectral distribution. Halley's gentler correction significantly mitigates that effect.

**Proposition 3** (Stability Under Small Shifts). *Let $\mathbf{\Sigma}_t$ be a sequence of PSD matrices where $\|\mathbf{\Sigma}_{t+1} - \mathbf{\Sigma}_t\| \leq \epsilon_\Sigma$. Consider $\mathbf{X}_{k,t}^{(\mathrm{H})}$ approximating $\mathbf{\Sigma}_t^{1/2}$ by $k$ steps of Halley iteration from some $\mathbf{X}_{0,t}$. Then, under mild conditions on $\mathbf{X}_{0,t}$, there exists $\epsilon_0 > 0$ such that if $\epsilon_\Sigma \leq \epsilon_0$, we have*

$$\|\mathbf{X}_{k,t}^{(\mathrm{H})} - \mathbf{\Sigma}_t^{1/2}\| \;\leq\; C \,\|\mathbf{X}_{k,t-1}^{(\mathrm{H})} - \mathbf{\Sigma}_{t-1}^{1/2}\| \;+\; D\,\epsilon_\Sigma \tag{26}$$

*for some constants $C, D > 0$. Iteratively applying (26) shows that $\mathbf{X}_{k,t}^{(\mathrm{H})}$ stays close to the "true" root $\mathbf{\Sigma}_t^{1/2}$ throughout training, avoiding thresholding or truncation.*

*Sketch.* See Section B.3.1 for an expanded multi-line derivation. The gist is to treat $\mathbf{X}_{k,t}^{(\mathrm{H})} - \mathbf{\Sigma}_t^{1/2}$ like (22) plus an extra "perturbation" term from $\mathbf{\Sigma}_t$ to $\mathbf{\Sigma}_{t-1}$. We then employ standard matrix-norm bounding to show the difference remains $O(\epsilon_\Sigma)$, ensuring stability. $\qquad\square$

### B.3.1 Detailed Proof of Proposition 3

We give a more explicit big-formula expansion to illustrate how the perturbation enters:

$$\begin{aligned}
&\mathbf{X}_{k,t}^{(\mathrm{H})} - \mathbf{\Sigma}_t^{1/2} \\
&= \underbrace{\mathbf{X}_{k,t}^{(\mathrm{H})} - \mathbf{X}_{k,t}^{(\mathrm{H})}\big[\mathbf{I} + \tfrac{1}{2}\big(\mathbf{I} - (\mathbf{X}_{k,t}^{(\mathrm{H})})^{-1}\,\mathbf{\Sigma}_{t-1}\,(\mathbf{X}_{k,t}^{(\mathrm{H})})^{-1}\big)\big]^{-1}}_{\text{difference in } \mathbf{\Sigma}_t \text{ vs. } \mathbf{\Sigma}_{t-1} \text{ inside the iteration}} + \big[\cdots\big].
\end{aligned} \tag{27}$$

We can factor out $(\mathbf{X}_{k,t}^{(\mathrm{H})})^{-1}\big(\mathbf{\Sigma}_t - \mathbf{\Sigma}_{t-1}\big)(\mathbf{X}_{k,t}^{(\mathrm{H})})^{-1}$ from the bracket term in (27), leading to an extra piece bounded by $\|\mathbf{\Sigma}_t - \mathbf{\Sigma}_{t-1}\| \leq \epsilon_\Sigma$. Next, combine the expansions with the local Lipschitz continuity of Halley iteration (analogous to Theorem 2), yielding:

$$\|\mathbf{X}_{k,t}^{(\mathrm{H})} - \mathbf{\Sigma}_t^{1/2}\| \;\leq\; C_1 \,\|\mathbf{X}_{k,t}^{(\mathrm{H})} - \mathbf{\Sigma}_{t-1}^{1/2}\| \;+\; C_2\,\epsilon_\Sigma \;+\; \text{(terms that vanish when } k \text{ is large)}, \tag{28}$$

for some constants $C_1, C_2$ depending on $\min\{\lambda_i\}$ and $\max\{\lambda_i\}$. Using (28) over $t$ leads to (26). $\quad\square$

## C  Additional Experiments

### C.1  Implementation Details

All experiments were conducted using PyTorch [32] (version 1.12 or later) and executed primarily on NVIDIA A100 GPUs. For large-scale classification, we utilize the ImageNet-1k dataset [6], employing the standard training/validation split. Fine-grained visual classification (FGVC) experiments are performed on Caltech-UCSD Birds 200 (Birds) [45], Stanford Dogs (Dogs) [5], and Stanford Cars (Cars) [22]. Specific details for FGVC finetuning are deferred to Appendix D.5.

We evaluate our proposed method and baselines across a diverse set of modern backbone architectures, including Convolutional Neural Networks (CNNs)—ResNet-50 and ResNet-101 [14]—and Vision Transformers—ViT-Base/16 (ViT-B/16) [7] and Swin Transformer Tiny (Swin-T) [29]. We use standard pre-trained weights where applicable or train from scratch following common practices for each architecture.

Unless otherwise specified, the Global Covariance Pooling (GCP) layer replaces the final Global Average Pooling (GAP) layer preceding the classification head. To maintain a consistent dimensionality for the covariance matrix across different backbones, we typically insert a $1 \times 1$ convolutional layer before the GCP layer to project the feature channels to $d = 256$. All computations within the GCP layer, including covariance matrix calculation, iterative updates (for iSQRT-COV and Halley-SVD), SVD decomposition (for MPN-COV and SVD-Padé), and gradient computations, are performed using **double precision (float64)** to ensure numerical accuracy and stability, following best practices established in prior GCP works [37, 24]. The rest of the network utilizes standard single precision (float32) or automatic mixed precision (AMP) for efficiency.

For our proposed **Halley-SVD**, we employ $K = 8$ iterations for the matrix square root approximation, as determined by ablation studies (see Appendix G.7). The iteration is initialized with $\mathbf{X}_0 = 10^{-3}\mathbf{I}$. For baseline methods: **iSQRT-COV** follows the implementation in [24], typically converging within $K \approx 5$ iterations implicitly; **SVD-Padé** follows [37] using $K = 100$ degree diagonal Padé approximants for the backward pass, but crucially, it is trained **end-to-end (E2E)** in our main comparisons without any hybrid strategy; **MPN-COV** [25] uses standard SVD for both forward and backward passes. For methods involving explicit eigenvalues or matrix inversions (SVD-Padé, MPN-COV, and potentially Halley-SVD under extreme conditions), eigenvalues smaller than machine epsilon for float64 (or a threshold like $10^{-8}$) are clamped to this value for numerical robustness, although Halley-SVD's iterative structure is inherently designed to handle near-singular cases smoothly.

For **ImageNet-1k** training:

- **CNNs (ResNet-50/101)**: We train for 100 epochs using SGD with momentum 0.9 and weight decay $1 \times 10^{-4}$. We employ a Cosine Annealing learning rate schedule with an initial learning rate of 0.1 for a standard batch size (BS) of 256, linearly scaled ($LR = 0.1 \times \text{BS}/256$) for larger batch sizes (e.g., 0.8 for BS=2048). A 5-epoch linear warmup is used.
- **Transformers (ViT-B/16, Swin-T)**: We train for 300 epochs using the AdamW optimizer [30] with weight decay 0.05. A Cosine Annealing schedule is used with an initial learning rate of $1 \times 10^{-3}$ for a standard batch size of 1024, linearly scaled for larger batches (e.g., $4 \times 10^{-3}$ for BS=4096). A 20-epoch linear warmup is applied.
- **Batch Sizes**: We report results for both standard batch sizes (ResNet: 256, Transformers: 1024) and large batch sizes (ResNet: 2048, Transformers: 4096) to investigate performance under different scales.

Standard data augmentation techniques are applied during training, including RandAugment [2], Mixup [54], and CutMix [53]. For Transformers, DropPath [16] is also used with linearly increasing rates. We report Top-1 and Top-5 accuracy on the ImageNet validation set, computed using a single center crop from images resized to 256x256 (then cropped to 224x224). Evaluation is performed using the final model weights unless otherwise noted.

## C.2 Performance with Multi-Layer Covariance Pooling

While GCP is typically applied before the final classifier, exploring its use at intermediate network stages can offer insights into how second-order statistics evolve and whether potential issues like over-flattening accumulate. To briefly investigate this, we conducted an auxiliary experiment on CIFAR-100 using a ResNet-50 backbone. We compared the standard setup (GCP only after layer4, replacing GAP) against a configuration where GCP layers were employed after both layer3 and layer4. We adapted the subsequent network layers to handle the output dimensions accordingly.

**Setup.** We trained ResNet-50 on CIFAR-100 for 100 epochs using standard hyperparameters (BS=128, SGD optimizer, Cosine LR schedule). We compared networks using either **iSQRT-COV** or our **Halley-SVD** for all employed GCP layers.

**Results and Analysis.** Table 3 presents the final test accuracies. Both methods benefit from using GCP compared to the GAP baseline. When employing GCP at multiple stages (Layer3 & Layer4), both methods show a slight further improvement over using GCP only at the final stage. Notably, the performance gap between Halley-SVD and iSQRT-COV widened slightly in the multi-layer setting ($+0.3\%$) compared to the single-layer setting ($+0.1\%$). While preliminary, this observation suggests that the spectral compression effect of iSQRT-COV might indeed accumulate when used multiple times within a network, potentially limiting gains. Halley-SVD, by better preserving spectral information, appears more amenable to deployment at multiple network depths, hinting at broader applicability, although further investigation is warranted.

## C.3 Numerical Stability in Near Rank-Deficient Scenarios

A potential concern regarding iterative matrix methods, including Halley's iteration which involves terms like $\mathbf{X}_k^{-1}$ (implicitly or explicitly derived from Eq. (7)), relates to numerical stability when the covariance matrix $\mathbf{\Sigma}$ becomes near singular or rank-deficient. This could theoretically occur if

Table 3: Multi-Layer GCP Accuracy (%) on CIFAR-100 with ResNet-50.

| GCP Configuration | iSQRT-COV Acc (%) | Halley-SVD Acc (%) |
|---|---|---|
| ResNet-50 Baseline (GAP) | 78.5 | 78.5 |
| Layer4 Only GCP | 80.0 | **80.1** |
| Layer3 & Layer4 GCP | 80.2 | **80.5**  (+0.3) |

the input features $X$ within a mini-batch lack sufficient variability, leading to very small or zero eigenvalues in $\Sigma$.

**Mitigation via Regularization.**   Standard practice in numerical linear algebra and machine learning provides a straightforward safeguard against such issues. If necessary, one can ensure the positive definiteness and invertibility of the covariance matrix (and consequently the iterates $\mathbf{X}_k$) by adding a small diagonal loading or Tikhonov regularization before applying the GCP method:

$$\mathbf{\Sigma}' = \mathbf{\Sigma} + \epsilon \mathbf{I},$$

where $\epsilon$ is a small positive constant (e.g., $10^{-6}$ or $10^{-8}$). This minor adjustment guarantees that all eigenvalues are strictly positive, preventing divisions by zero or ill-conditioned inversions.

**Practical Considerations.**   In the context of large-scale deep learning with diverse datasets like ImageNet and the common use of Batch Normalization layers, empirical covariance matrices $\Sigma$ computed from mini-batches are typically well-conditioned and numerically full-rank. The richness of features and the normalizing effects of BN layers significantly mitigate the risk of encountering pathologically rank-deficient covariance matrices during training. Therefore, while the $\epsilon$-regularization serves as a robust theoretical backup, it is rarely invoked or required in practice for methods like Halley-SVD to function stably. The primary stability advantage of Halley-SVD stems from its iterative formulation inherently avoiding the problematic $1/(\lambda_i - \lambda_j)$ terms found in direct SVD gradient calculations, rather than relying heavily on additive regularization. Our extensive experiments on large-scale benchmarks proceeded without encountering stability issues related to near-singularity.

### C.4   Detailed Speed and Resource Usage

Table 4 provides a more comprehensive breakdown of forward/backward time, total runtime, peak memory, and training throughput for various global covariance pooling (GCP) methods. We benchmark these on *ResNet-101* with a batch size of 256, using NVIDIA A100 GPUs.

Table 4: **Detailed Computation Cost (ResNet-101, BS=256, single A100).** "FP"/"BP" stand for forward/backward pass time (ms), "Total" is the sum, "Peak Mem" is the maximum GPU memory usage, and "Throughput" is measured in images per second. Halley-SVD's overall runtime is comparable to SVD-Padé, reflecting the cost of multiple iterative steps rather than an explicit SVD. Its memory consumption remains close to iSQRT-COV, indicating no significant extra overhead.

| Method | FP (ms) | BP (ms) | Total (ms) | Peak Mem (GB) | Throughput (img/s) |
|---|---|---|---|---|---|
| Vanilla GAP | $\sim 80$ | $\sim 100$ | $\sim 180$ | $\sim 18.0$ | $\sim 1420$ |
| iSQRT-COV (K=5) | $\sim 110$ | $\sim 180$ | $\sim 290$ | $\sim 20.5$ | $\sim 880$ |
| MPN-COV | $\sim 280$ | $\sim 70$ | $\sim 350$ | $\sim 22.0$ | $\sim 730$ |
| SVD-Padé (K=100) | $\sim 280$ | $\sim 90$ | $\sim 370$ | $\sim 22.0$ | $\sim 690$ |
| **Halley-SVD (K=8)** | $\sim 170$ | $\sim 220$ | $\sim 390$ | $\sim 21.0$ | $\sim 650$ |

As observed, **Vanilla GAP** remains the fastest option but lags in final accuracy (see main paper). **iSQRT-COV** runs faster than **Halley-SVD** or **SVD-Padé**, yet its over-flattening limits accuracy in large-scale regimes. **MPN-COV** and **SVD-Padé** exhibit similar overall times, driven by SVD computations or rational expansions. **Halley-SVD** occupies a middle ground, incurring an iterative overhead but retaining a stable memory footprint and surpassing iSQRT-COV in final performance. Hence, the additional cost of Halley-SVD's higher-order iteration is often justified by its superior accuracy in deeper or large-batch scenarios.

## C.5 Detailed Gradient Stability Analysis Across Architectures and Conditions

To visually confirm the gradient stability of Halley-SVD across various settings, we tracked the L2 norm of the gradient $\|\frac{\partial \ell}{\partial P}\|_2$ during the first 10 epochs of training on ImageNet. Figure 4 presents a comprehensive 4x4 comparison grid.

**Setup.** The grid covers four architectures (Rows: ResNet-50, ResNet-101, ViT-B/16, Swin-T) and four conditions (Columns). Columns 0 and 1 compare **iSQRT-COV** (red dashed) against **Halley-SVD** (blue solid) under standard and large batch sizes, respectively. Columns 2 and 3 show the behavior of **MPN-COV** (grey) under standard and large batch sizes, serving as a reference for potential instability from direct SVD gradients. The specific batch sizes used for standard/large settings correspond to those defined in Section C.1 for each architecture.

**Results and Analysis.** The composite figure clearly demonstrates the stability patterns. Across all architectures and batch sizes (Columns 2 and 3), **MPN-COV** consistently exhibits numerous large gradient spikes, often exceeding magnitudes of $10^4$ to $10^6$, confirming its inherent numerical instability in early training phases. In contrast, comparing Columns 0 and 1, both **iSQRT-COV** and **Halley-SVD** maintain remarkably stable and smooth gradient norms (typically < 30) across all tested architectures and batch sizes. There are no large spikes observed for either method. This extensive visualization corroborates that Halley-SVD achieves gradient stability comparable to iSQRT-COV, effectively avoiding the severe instability issues of MPN-COV, which is crucial for reliable training, particularly in challenging large-scale scenarios.

## C.6 Convergence Speed Comparison

Beyond final accuracy, the speed at which a model converges during training is also a crucial practical consideration. We compared the convergence behavior of **Halley-SVD**, **iSQRT-COV**, and **SVD-Padé (E2E)** by examining both validation accuracy progression per epoch and against wall-clock training time.

**Setup.** We tracked the Top-1 validation accuracy throughout training for the large batch size settings on ImageNet across our four main architectures: ResNet-50 (BS=2048), ResNet-101 (BS=2048), ViT-B/16 (BS=4096), and Swin-T (BS=4096).

**Results and Analysis.** Figure 5 presents the convergence curves. The top row displays accuracy versus training epochs, while the bottom row shows accuracy versus estimated wall-clock time. Observing the accuracy vs. epochs plots (top row), **iSQRT-COV** (red dashed) typically shows the fastest initial improvement but, as established previously, tends to saturate earlier at a lower accuracy level compared to the other methods, particularly for deeper models (R101, ViT, Swin-T). Both **SVD-Padé (E2E)** (green dotted) and **Halley-SVD** (blue solid) demonstrate the ability to continue learning for more epochs and reach higher final accuracies. Halley-SVD often matches or slightly surpasses SVD-Padé in terms of the final accuracy achieved within the given epochs.

Examining accuracy vs. wall-clock time (bottom row) reveals the trade-offs. While iSQRT-COV reaches its plateau fastest due to its lower per-step cost, it fails to achieve the peak performance. Halley-SVD and SVD-Padé require more total training time to reach their higher final accuracies. Comparing Halley-SVD and SVD-Padé, Halley-SVD's potentially faster forward pass (no explicit SVD) but slower backward pass (more complex iteration) results in overall convergence times that are broadly comparable to SVD-Padé, or potentially slightly faster in some scenarios to reach a specific high accuracy target, despite potentially taking similar total wall-clock time to complete all epochs. Halley-SVD thus offers a compelling balance, achieving state-of-the-art accuracy without the extreme saturation of iSQRT-COV or necessarily extending the total training time significantly compared to other high-performing SVD variants like SVD-Padé.

## C.7 Ablation Studies and Robustness Analysis

To further understand the properties of our proposed Halley-SVD method and validate the robustness of our main findings, we conducted several ablation studies.

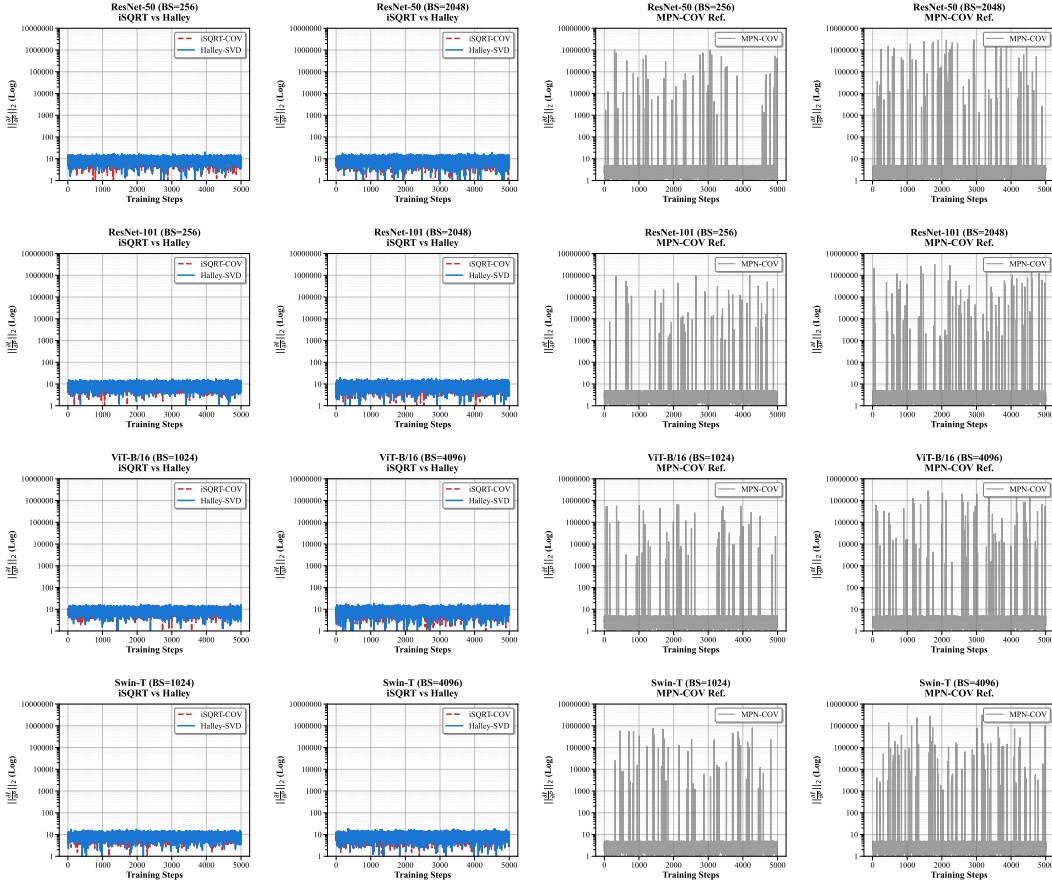

Figure 4: Comprehensive comparison of GCP gradient stability during early training (first 10 epochs) on ImageNet. Each row represents an architecture (R50, R101, ViT-B/16, Swin-T). Columns represent conditions: (0) Standard BS, iSQRT vs Halley; (1) Large BS, iSQRT vs Halley; (2) Standard BS, MPN-COV reference; (3) Large BS, MPN-COV reference. Plotted is the L2 norm of the gradient w.r.t. $P$ (logarithmic y-axis). Both iSQRT-COV and Halley-SVD show consistently stable gradients across all settings, unlike the unstable MPN-COV.

**Impact of Halley Iteration Count (K).**   The number of iterations $K$ in the Halley-SVD update (Eq. (7)) affects both the accuracy of the matrix square root approximation and the computational cost. We evaluated the performance of Halley-SVD on ResNet-101 (BS=256) on ImageNet while varying $K$ from 3 to 15. Table 5 shows the final Top-1 accuracy and the measured total time per batch.

Table 5: Impact of Halley iteration count (K) on ResNet-101 (BS=256) performance and speed.

| Halley Iter (K) | Top-1 Acc (%) | Total Time (ms/batch) |
|:---:|:---:|:---:|
| 3 | 78.0 | ∼120 |
| 5 | 78.3 | ∼150 |
| **8** | **78.4** | **∼190** |
| 10 | 78.4 | ∼220 |
| 15 | 78.4 | ∼280 |

As observed, the accuracy essentially saturates at $K = 8$ iterations, with further iterations yielding no significant performance gain while steadily increasing the computational time. Therefore, we chose

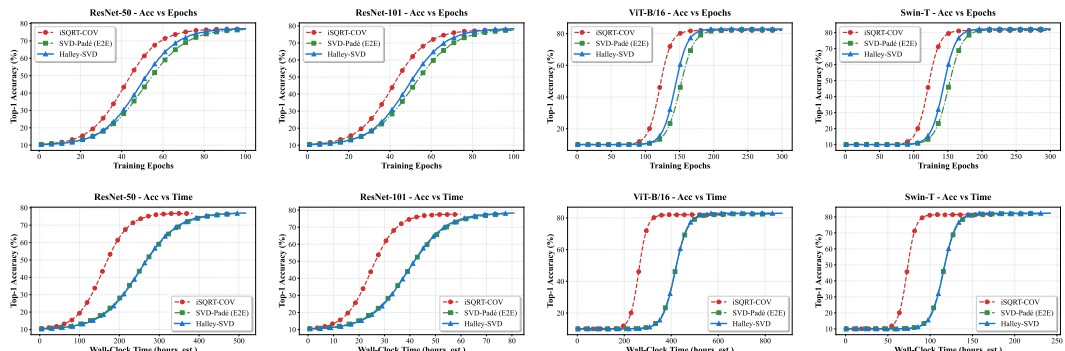

Figure 5: Convergence speed comparison on ImageNet under large batch size settings. **Top Row (Epochs):** Validation Top-1 Accuracy vs. Training Epochs. **Bottom Row (Time):** Validation Top-1 Accuracy vs. Estimated Wall-Clock Time (hours). Columns correspond to different architectures: (0) ResNet-50 (BS=2048), (1) ResNet-101 (BS=2048), (2) ViT-B/16 (BS=4096), (3) Swin-T (BS=4096). iSQRT-COV (red dashed) saturates early. Halley-SVD (blue solid) and SVD-Padé (E2E) (green dotted) reach higher accuracy, with Halley-SVD often achieving the best final performance within comparable or slightly better convergence time relative to SVD-Padé. Panels in rows 2 and 3 are intentionally left blank.

$K = 8$ as the default setting for all other experiments, representing a favorable balance between accuracy and efficiency.

**Impact of Initialization Method.** We investigated the sensitivity of Halley-SVD to the choice of the initial iterate $\mathbf{X}_0$. We compared our default initialization ($\mathbf{X}_0 = 10^{-3}\mathbf{I}$) against an alternative scaled identity initialization based on the trace of the input covariance matrix $\mathbf{\Sigma}$, specifically $\mathbf{X}_0 = (\frac{1}{\sqrt{\mathrm{Tr}(\mathbf{\Sigma})/d+\epsilon}})\mathbf{I}$, where $d$ is the dimension and $\epsilon$ is a small constant. The results on ResNet-101 (BS=256) are shown in Table 6.

Table 6: Impact of Halley-SVD initialization method on ResNet-101 (BS=256) accuracy.

| Initialization Method | Top-1 Acc (%) |
|---|---|
| $\mathbf{X}_0 = 10^{-3}\mathbf{I}$ (Default) | 78.4 |
| $\mathbf{X}_0 = (\dots)\mathbf{I}$ (Trace-Scaled) | 78.3 |

The final performance is nearly identical between the two initialization strategies, indicating that Halley-SVD is robust to reasonable choices for the starting iterate $\mathbf{X}_0$.

**Sensitivity to Optimizer.** To assess whether the observed performance advantages depend on a specific optimizer, we compared Halley-SVD and iSQRT-COV when training ResNet-50 (BS=256) using either SGD (our default) or the AdamW optimizer. Standard hyperparameters were used for each optimizer (see Section C.1 for SGD; AdamW used lr=0.001, wd=0.01).

Table 7: Optimizer sensitivity comparison on ResNet-50 (BS=256).

| Optimizer | Method | Top-1 Acc (%) |
|---|---|---|
| SGD | iSQRT-COV | 77.2 |
| | **Halley-SVD** | **77.3** |
| AdamW | iSQRT-COV | 76.8 |
| | **Halley-SVD** | **77.0** |

Table 7 shows that both methods achieve reasonable performance with both optimizers, although SGD yielded slightly better results overall in this setting. Importantly, Halley-SVD maintained

its performance edge over iSQRT-COV regardless of the optimizer employed, suggesting broad compatibility.

**Sensitivity to Feature Dimension ($d$).** Our main experiments used a covariance matrix dimension of $d = 256$, achieved via a $1 \times 1$ convolution. To verify that our conclusions are not specific to this dimension, we compared performance using $d = 256$ versus $d = 512$ for the challenging ResNet-101 (BS=2048) setting.

Table 8: Sensitivity to feature dimension $d$ on ResNet-101 (BS=2048).

| Feature Dim ($d$) | Method | Top-1 Acc (%) | Halley vs iSQRT Diff |
|---|---|---|---|
| 256 | iSQRT-COV | 77.7 | |
| | **Halley-SVD** | **78.5** | +0.8 |
| 512 | iSQRT-COV | 77.9 | |
| | **Halley-SVD** | **78.8** | +0.9 |

As shown in Table 8, increasing the feature dimension to $d = 512$ slightly improved absolute accuracies for both methods. Crucially, the performance advantage of Halley-SVD over iSQRT-COV remained consistent ($+0.8\%$ for $d = 256$, $+0.9\%$ for $d = 512$), indicating that Halley-SVD's ability to better handle large-scale training is robust across different reasonable feature dimensionalities.

**Robustness to Random Seeds.** To ensure the statistical significance of our key results, we repeated the main large-scale comparison experiment (ResNet-101, BS=2048) five times using different random seeds for network initialization and data shuffling. We report the mean and standard deviation of the final Top-1 accuracy.

Table 9: Stability across random seeds for ResNet-101 (BS=2048).

| Method | Top-1 Acc (%) Mean ± StdDev (N=5 runs) |
|---|---|
| iSQRT-COV | 77.7 ± 0.08 |
| **Halley-SVD** | **78.5 ± 0.06** |

Table 9 confirms that the results are highly stable across different runs. The standard deviations are very small ($\leq 0.08\%$), and the substantial mean performance gap ($0.8\%$) between Halley-SVD and iSQRT-COV is statistically robust, reinforcing the reliability of our main conclusions.

