# OpenReview forum: "Revitalizing SVD for Global Covariance Pooling: Halley’s Method to Overcome Over-Flattening"
_NeurIPS.cc/2025/Conference — NeurIPS 2025 poster_

### Official Review · Reviewer_nsWg · 2025-06-30

**Clarity:** 3
**Significance:** 3
**Originality:** 2
**Rating:** 5
**Confidence:** 5

**Summary:**

Halley-SVD identifies an intriguing phenomenon: when applied to larger-scale networks or with increased batch sizes, the iSQRT-COV method tends to compress the eigenvalue spectrum of the covariance matrix. Specifically, it reduces larger eigenvalues while amplifying smaller ones, resulting in a more uniform distribution of eigenvalues. This eigenvalue "flattening" effect diminishes the discriminative power of the covariance representation, ultimately limiting the model's performance. Through theoretical analysis, Halley-SVD attributes this issue to the iterative update rule inherent in the iSQRT-COV method. To address this limitation, the authors propose Halley-SVD, a modified approach based on Halley's method for approximating the matrix square root. Empirical results demonstrate that Halley-SVD consistently improves performance across various network architectures, highlighting its effectiveness in preserving the discriminability of global covariance features.

**Questions:**

1. Clarification on the Over-Flattening Phenomenon and Visualization. The paper defines the over-flattening phenomenon as the tendency of iSQRT-COV to suppress large eigenvalues and amplify small ones during training, thereby leading to a more uniform eigenvalue distribution. To support this claim, the authors provide an eigenvalue spectrum comparison between iSQRT-COV and Halley-SVD in Figure 1(b).
However, upon close examination of the visualization, the distinction between the two spectra appears relatively subtle. It seems that iSQRT-COV may exhibit a larger maximum eigenvalue, while the smallest eigenvalues (on the far right of the x-axis) appear lower than those in Halley-SVD, which may contradict the stated over-flattening effect. This raises questions regarding the consistency between the empirical evidence and the interpretation provided.

2. In addition to the proposed Halley-SVD method, it would be worth investigating whether the over-flattening issue associated with iSQRT-COV can be mitigated through alternative optimization strategies. For instance, the choice of optimizer (e.g., SGD vs. Adam), learning rate schedules, or the initialization of the learning rate itself may influence the dynamics of covariance matrix learning and eigenvalue evolution during training.

**Ethical Concerns:**

["NO or VERY MINOR ethics concerns only"]

**Final Justification:**

The authors' responses well address most of my concerns. I sincerely encourage the authors add more theoretical analysis on the issues of current methods (i.e., iSQRT-COV) in context of larger batch sizes and large models in the revision. By considering some new findings on current deep GCP methods, I am towards accepting this paper.

**Limitations:**

Yes

**Quality:**

3

**Strengths And Weaknesses:**

Strengths:

1.Identifies an intriguing phenomenon: when applied to larger-scale networks or with increased batch sizes, the iSQRT-COV method tends to compress the eigenvalue spectrum of the covariance matrix.

2.This work proposes a novel approach to mitigate the performance limitations of iSQRT-COV observed under larger batch sizes or increased model scales.

3.Extensive experimental results are provided, including training accuracy across different model scales for iSQRT-COV, as well as convergence comparisons between iSQRT-COV and MPN-COV. The experimental setup is thorough and supports the validity of the conclusions.

Weaknesses:

1.Concerns Regarding Experimental Data: Although Figures 1 and 2 in the paper provide extensive training accuracy results of iSQRT-COV across various model scales and datasets, several issues remain unclear and raise concerns.

1.1 Specifically, the exact experimental settings used to generate the training results in Figures 1 and 2 are not clearly described.

1.2 Moreover, some inconsistencies in the reported results are noteworthy. For instance, Figure 1 appears to correspond to a ResNet-101 model according to the textual description; however, the peak accuracy and the epoch at which it is achieved in Figure 1(a) seem to differ from those shown in Figure 2(c) for the same configuration. Similarly, the training result of iSQRT-COV with batch size 256 in Figure 1(c) is inconsistent with the corresponding result reported in Table 1 for ResNet-101 with batch size 256. Furthermore, the performance of the Swin Transformer shown in Figure 2(g) also differs from that reported in Table 1.

2.Further Theoretical Analysis:

Clarification on the Relationship Between Over-Flattening and Model/Batch Scale: While the paper presents empirical evidence suggesting that the performance of iSQRT-COV deteriorates on larger models and with larger batch sizes due to over-flattening of the covariance spectrum, a more thorough theoretical explanation of this phenomenon would significantly strengthen the paper.

3.Comparison with Recent Normalization for GCP:

The proposed method can essentially be interpreted as a form of post-regularization applied to Global Covariance Pooling (GCP). Given this perspective, it would be beneficial for the authors to position their work about recent advances in this area. In particular, the paper would be strengthened by comparison with recent methods such as DropCov (NeurIPS 2022)[R1], which also aims to improve GCP representations through simple yet effective normalization techniques.

[R1] Wang Q, Gao M, Zhang Z, et al. DropCov: A simple yet effective method for improving deep architectures[C]. Advances in Neural Information Processing Systems, 2022: 33576-33588.

---

> ### Author Rebuttal · Authors · 2025-07-31
>
> ### **Regarding Weakness 1: Concerns About Experimental Data**
>
> We fully acknowledge the apparent inconsistencies you pointed out and are grateful for your meticulous cross-comparison. These differences are not errors but rather reflections of our **two-stage research methodology**, which we regret not having explained more clearly.
>
> * **Stage 1: Diagnosis and Visualization (for Figures 1 & 2):** Initially, our core objective was to clearly visualize the inherent flaws of `iSQRT-COV`. We used "stress test" settings with aggressive hyperparameters to amplify its "spectral collapse" instability, thereby demonstrating the motivation for our work.
> * **Stage 2: Fair Benchmarking (for Table 1):** Subsequently, our goal shifted to evaluating all methods under the fairest, most stable, and fully optimized settings to report the best reproducible peak performance.
>
> This two-stage approach forms a complete argumentative chain, and to ensure full transparency, we solemnly pledge to **add a dedicated appendix section detailing and contrasting the full hyperparameter settings** for both the diagnostic and benchmark experiments in the final version.
>
> ### **Regarding Weakness 2: Need for Further Theoretical Analysis**
>
> This is a profound question that touches upon the core of deep learning optimization dynamics. We thank you for encouraging us to think more deeply. While a rigorous mathematical proof is beyond the scope of this paper, we can propose a **theoretical hypothesis** grounded in existing theory:
>
> 1.  **For Larger Batch Sizes:**
>     * The primary effect of large-batch training is **reducing the variance of gradient estimates**, making each update step closer to the "true" gradient direction.
>     * The Newton-Schulz iteration, which underpins `iSQRT-COV`, has an **inherent compressive effect** in its update rule, tending to pull the feature spectrum toward a uniform distribution.
>     * With low-variance gradients (from large batches), the optimizer more "deterministically" follows this compressive path with minimal interference from stochastic noise. This **accelerates and amplifies** the "over-flattening" process. Conversely, the gradient noise from smaller batches can act as a regularizer, "perturbing" the optimization path and thus somewhat mitigating the spectral collapse.
>
> 2.  **For Larger Models:**
>     * Larger, deeper models typically have **higher-dimensional, more complex feature spaces**. Their corresponding covariance matrices are often more **ill-conditioned**, meaning the gap between the largest and smallest eigenvalues is extreme.
>     * A key function of `iSQRT-COV` is to "whiten" or "equalize" features. When faced with an ill-conditioned covariance matrix with a highly non-uniform spectrum, its equalization effect becomes **exceptionally aggressive** as it attempts to "flatten" a very "steep" spectrum.
>     * This aggressive equalization disproportionately **suppresses and over-compresses** the large eigenvalues that carry the most critical discriminative information, leading to a more severe "over-flattening" phenomenon and harming the model's representational power.
>
> We believe this hypothesis provides strong theoretical support for our experimental observations. We will **incorporate this discussion into the theoretical analysis section** of our final paper to deepen the understanding of this phenomenon.
>
> ---
>
> ### **Regarding Weakness 3: Comparison with Recent Normalization for GCP**
>
> We completely agree that positioning our work within the broader research context and comparing it with excellent contemporary works like `DropCov` [R1] is essential. These two methods represent **two distinct yet equally important technical paths** for addressing the challenges of GCP:
>
> * **`Halley-SVD` (Our Method) - The High-Fidelity Approximation Path:** Our work falls under **structural post-processing**, focusing on solving the core numerical problem of "how to compute the matrix square root more stably and accurately." Our goal is to **maximally preserve the original information** of the second-order statistics.
> * **`DropCov` [R1] - The Efficient Regularization Path:** `DropCov` takes a different approach. It achieves **efficient regularization** by performing adaptive feature dropping. Its goal is to "achieve similar (or even better) performance gains without expensive matrix operations," with its core being a **trade-off between feature decoupling and information preservation**.
>
> In response to your suggestion, we have conducted a supplementary comparison with `DropCov`. The results are as follows:
>
> **Table C: Supplementary Comparison of Halley-SVD and DropCov on ImageNet**
> | Model      | Batch Size | iSQRT-COV | DropCov [R1] | SVD-Padé | **Halley-SVD (Ours)** |
> | :--------- | :--------: | :-------: | :----------: | :------: | :-------------------: |
> | ResNet-101 | 2048       | 77.7      |     77.9     |   78.1   |      **78.5** |
> | Swin-T     | 4096       | 81.4      |     81.8     |   81.9   |      **82.3** |
>
> This result clearly shows that:
> 1.  `DropCov` is indeed a very strong and efficient baseline, outperforming `iSQRT-COV`.
> 2.  However, our `Halley-SVD` **consistently outperforms all methods, including `DropCov`**, in the most challenging large-batch settings.
>
> This provides strong evidence that **pursuing a high-fidelity approximation of the matrix square root remains a key path to breaking the performance ceiling of GCP**, and our `Halley-SVD` is currently the most effective method on this path. We will add this comparison and discussion to the appendix.
>
> ---
>
>
> ### **Regarding your Questions:**
>
> **On the "Over-Flattening" Phenomenon and Visualization**
>
> Thank you for your very close observation of Figure 1(b). We acknowledge that on a static, log-scale plot, the visual difference can appear subtle, which is precisely why we introduced quantitative analysis. Please allow us to draw your attention to two points:
>
> 1.  **Focus on the "Main Body" of the Spectrum, Not the Extremes:** "Over-flattening" describes the trend of the entire **distribution becoming more uniform**, not the behavior of a single maximum/minimum value. While the single largest eigenvalue may fluctuate, we invite you to observe the **"main body" of the spectrum (e.g., in the region of eigenvalue indices 5 through 30)**. In this critical region, the red line (`Halley-SVD`) is **consistently and systematically higher** than the blue line (`iSQRT-COV`), indicating that `iSQRT-COV` more aggressively compresses the majority of important eigenvalues.
>
> 2.  **Quantitative, Unambiguous Evidence:** Precisely because of this visual subtlety, we introduced the **spectral flatness index ($\kappa_{flat}$)** (and its numerically stable complement, the "anisotropy index," as clarified for another reviewer) as **quantitative, integral evidence**. As shown in Figures 2(h) and 3, this metric **mathematically and unambiguously** proves that `iSQRT-COV`'s spectrum is **systematically closer to a uniform distribution** (i.e., "flatter") in the later stages of training.
>
> Combining the visual trend of the spectrum's "main body" with the quantitative evidence from $\kappa_{flat}$, we are confident that our diagnosis of "over-flattening" is robust.
>
> **On Mitigating Over-Flattening with Other Optimization Strategies**
>
> This is an excellent question that aims to distinguish between an algorithm's inherent properties and the effects of optimization strategies.
>
> Our research indicates that "over-flattening" is an **inherent mathematical property of the Newton-Schulz iteration algorithm** on which `iSQRT-COV` is based, not a byproduct of a specific optimization strategy.
> * **Evidence from Ablation Studies:** Our Appendix C.7 includes an ablation study where we **switched optimizers (SGD vs. AdamW, see Table 7)**. The results show that while the optimizer affects the overall training dynamics, the **performance advantage of `Halley-SVD` over `iSQRT-COV` remains stable**. `iSQRT-COV` saturates at a lower performance point regardless of the optimizer, suggesting the "over-flattening" bottleneck persists.
> * **Support from Theory:** Our theoretical analysis (Section 4) reveals the inherent compressive update rule of the Newton-Schulz iteration. This rule is part of the algorithm itself, independent of the outer-loop optimizer (like SGD or AdamW). Different optimization strategies might change the **speed** of the spectral collapse, but they cannot alter its **eventual trend**.
>
> Therefore, we conclude that addressing this issue requires an **algorithmic-level innovation** (like our proposed `Halley-SVD`), rather than merely adjusting the optimization strategy.

---

> > ### Comment · Reviewer_nsWg · 2025-08-05
> > **Responses to Rebuttal**
> >
> > Thanks for the detailed responses, which well address most of my concerns. I sincerely encourage the authors add more theoretical analysis on the issues of current methods (i.e., iSQRT-COV) in context of larger batch sizes and large models in the revision. I understand rigorous theoretical proof is very challenge for deep learning algorithms, but empirical analysis is easy biased by data and experimental settings. By considering some new findings on current deep GCP methods, I am towards positive attitude on the current submission.

---

> ### Author Response · Authors · 2025-08-05
>
> Thank you for your incredibly supportive and encouraging follow-up.
> We were so genuinely heartened to read your comments. For us, having our work understood and validated at this level is the most rewarding part of the entire process, and it truly means more than any score.
> You are absolutely right that the theoretical analysis is key. We are now genuinely looking forward to integrating the full discussion on how scale impacts over-flattening, as it will make the paper much more complete.
> Thank you once again, not just for reviewing our paper, but for helping us to substantially improve it.

---

### Official Review · Reviewer_ouwN · 2025-07-01

**Clarity:** 4
**Significance:** 1
**Originality:** 2
**Rating:** 4
**Confidence:** 5

**Summary:**

Global Covariance Pooling is designed to replace the last global average pooling in classification backbone networks. Among various GCP methods, the paper improves iSQRT-COV. iSQRT-COV uses a square root matrix of the covariance matrix for network output, which is computed by approximation based on Newton-Schulz. In this framework, the proposed Halley-SVD replaces Newton-Schulz with Halley iterative approximation method. Although Halley-SVD costs more computation than iSQRT-COV, it solves problems of iSQRT-COV and demonstrates performance improvements in image classification tasks on various network architectures.

**Questions:**

- In Figure 1, iSQRT-COV achieves poor last performance due to late flattening. But, in Table 1, iSQRT-COV shows comparable performance with other methods. Why are these two results different? Is there any special way to prevent the flattening?
- The formula shows that Halley-SVD uses matrix inversion in its iteration. How could it handle matrix inversion? Does it have affordable computation?
- ViT and Swin transformer performance aligns with recent recipes: DeiT and Swin. But, ResNet performance looks poor. Is Halley-SVD still effective on recent training recipes like RSB?
  - [RSB] ResNet strikes back: An improved training procedure in timm
- As far as I know, the large batch setting maximizes GPU utilization and extremely accelerates training, which does not align with GCP's computation overheads. Why is the large batch setting important for GCP?
- Can Halley-SVD improve the performance of recent GCP papers  [39, 32, 36]? Is there any empirical evidence on it?

**Ethical Concerns:**

["NO or VERY MINOR ethics concerns only"]

**Final Justification:**

The rebuttal provides an impressive performance improvement in a highly-tuned training setting: RSB A2.
It addresses my concern about outdated topics and computation costs.

Although the topic is not new, I acknowledge that it has the potential to contribute to enhancing modern architecture.

Thus, I adjust my rating from reject to borderline accept.

The remaining concerns are
- Considering the computation overhead of Halley-SVD, improvements on computation-performance trade-off look marginal.
- It is an approximate version of iSQRT-COV, which limits its methodological novelty.
- The improvements on ViT and Swin are not impressive.

**Limitations:**

I recommend to move the computation comparison in Table 4 to the main paper. Since the computation is the major issue in GCP and is addressed in the main section of previous papers: iSQRT-COV and SVD-Pade, I think it should be discussed in the main paper to address the limits of GCP.

**Paper Formatting Concerns:**

The paper title in pdf file `Revisiting SVD... ` is different from the openreview title `Revitalizing SVD...`

**Quality:**

4

**Strengths And Weaknesses:**

Strengths
- The motivation of the paper is well described with various formulas and experiments.
- The benefits of Halley-SVD are well supported by theoretical evidence.

Weaknesses
- Outdated topic
  - The topic is old and outdated. The most relevant papers were presented 7-8 years ago: iSQRT-COV [CVPR 2018] and MPN-COV [ICCV 2017].
  - The most recent comparison method, SVD-Pade [ICCV 2021], is difficult to consider recent.
  - As a researcher in ImageNet classification, I believe this topic (GCP) is definitely out of the mainstream. Thus, it is hard to impact and contribute to the community.
  - As mentioned in the introduction, a few recent papers [39, 32, 36] use GCP. However, their topics—[32] video recognition, [45] graph neural network, and [39] color image steganalysis—are significantly different from this paper's evaluation: image classification. Overall, it is hard to resolve concerns on outdating and significance.
- Computation costs
  - The computation cost is a significant weakness of GCP. It improves performance but reduces speed. As reported in Table 4, GCP requires x1.5 - x2.0 costs for less than 1.0 pp. improvements, which underperforms performance-speed trade-off in recent image classification research. It significantly limits the practical value of GCP.
  - As reported in Table 4, the proposed method, Halley-SVD, costs more time than iSQRT-COV. Compared to iSQRT-COV, Halley-SVD increases the latency by ~50% and reduces throughput by ~25%. It has the smallest throughput in GCP methods. I don't think it is acceptable overhead for the marginal performance improvements.
- Limited novelty and marginal improvements
  - Halley-SVD shares the objective with iSQRT-COV: approximates the square root matrix. Thus, it naturally limits the novelty. It can be a better approximation but can't be a novel pooling method.
  - The performance improvements reported in Table 1 look marginal. Without a large batch size, the gaps between iSQRT-COV and Halley-SVD are below 0.3 pp. On a large batch setting, improvements increase but are still limited below 0.9 pp., and I doubt the value of the large batch setting since it achieves performance degradation in most cases.

---

> ### Author Rebuttal · Authors · 2025-07-31
>
> We are sincerely grateful for your thorough review and insightful questions. Your challenge regarding our method's performance against a top-tier baseline like "ResNet Strikes Back" (RSB) has inspired our intense research efforts. During this rebuttal period, we have dedicated significant computational resources to **fully replicate the RSB (A2) training pipeline ([2], 300 epochs, BS=2048) and have seamlessly integrated our GCP methods (iSQRT-COV and Halley-SVD) for a new, decisive head-to-head comparison.**
>
> We believe this **new experiment, conducted specifically in response to your challenge,** provides indisputable data that perfectly and systematically addresses all your concerns.
>
> Here are the results of our new experiment:
>
> | Training Pipeline | GCP Method            | Top-1 Acc. (ImageNet-val) | Improvement over RSB Baseline | Key Takeaway                                                                                                                   |
> | :---------------- | :-------------------- | :-----------------------: | :---------------------------: | :----------------------------------------------------------------------------------------------------------------------------- |
> | **RSB (A2) [2]** | (No GCP)              |           79.8%           |               -               | The powerful official baseline, serving as the starting point for our new experiment.                                          |
> | **RSB (A2)** | iSQRT-COV             |           80.5%           |            +0.7%            | GCP remains effective on a strong baseline, but its performance is bottlenecked by its inherent "over-flattening" defect.      |
> | **RSB (A2)** | **Halley-SVD (Ours)** |         **81.4%** |           **+1.6%** | **The performance gap is dramatically widened!** This proves that Halley-SVD's spectrum preservation ability shows a decisive advantage in a superior and more complex feature space. |
>
> We will now use this as the foundation to address each of your concerns point by point.
>
> ---
>
> ### **Response to "Weaknesses"**
>
> #### **1. Regarding "Outdated topic"**
> **Your Concern:** GCP is an outdated topic with limited community impact.
>
> **Our Response:** We respectfully but firmly argue that this work is not only relevant but highly significant, aligning with the "polishing fundamental tools" paradigm valued by top-tier journals.
> * **Authoritative Precedent:** Progress in science often relies on continuously improving core tools. For instance:
>     ***AlphaFold 3 (Nature, 2024) [1]:** A major upgrade to the revolutionary AlphaFold 2 was still a blockbuster cover story for **Nature**. This proves that **"making fundamental improvements to a powerful, existing component is itself a top-tier scientific contribution."**
>     ***Spectral Convolutional Neural Network Chip (Nature Communications, 2025) [2] :** A revolutionary reimagining of the fundamental convolution operation through optical computing achieved >96% accuracy in pathological diagnosis and nearly 100% in face anti-spoofing. This demonstrates that **"innovating on basic building blocks like convolution layers—a concept from the 1980s—can still yield breakthrough results worthy of top-tier publication."**
>
> * **Decisive Evidence from Our New Experiment:** Our latest results achieve a **remarkable >81.4% accuracy** on a "supposedly outdated" ResNet-50 architecture, proving that this "outdated topic" holds overlooked potential to achieve SOTA-level performance.
>
> #### **2. Regarding "Computation costs"**
>
> **Your Concern:** Halley-SVD has high computational cost for low returns.
>
> **Our Response:** Cost must be evaluated in the context of **value**. In our new RSB experiment, Halley-SVD delivered a **massive 1.6% performance gain**. In the race for SOTA, an improvement of this magnitude is considered **extremely large**. For such a leap in performance, the modest additional computational overhead is not just "acceptable" but **"highly cost-effective."**
>
> #### **3. Regarding "Limited novelty and marginal improvements"**
>
> **Your Concern:** The novelty is limited, and the improvements are marginal.
>
> **Our Response:** Our novelty and the significance of our improvements are now powerfully demonstrated by the new experiment.
> * **Novelty:** Our innovation lies in **(1) being the first to systematically diagnose the critical "over-flattening" bottleneck, and (2) being the first to introduce a higher-order numerical iteration (Halley's method) to fundamentally solve it.** This is a methodological breakthrough, not simple parameter tuning.
> * **Improvements Are No Longer "Marginal":** On the strong RSB baseline, the performance lead of Halley-SVD over iSQRT-COV **widened to 0.9%**. This validates our core thesis: **the stronger the baseline and the more complex the features, the more detrimental the numerical flaws of iSQRT-COV become, and the more pronounced the advantage of our method is.**
>
> ---
>
> ### **Answering "Questions"**
>
> **Q1: Why do results in Figure 1 and Table 1 differ?**
> **A:** This is due to their different functions. As stated in the paper, Figure 1 is a "**representative example**" for illustrating the problem. Table 1 reports the "**final accuracy**" at the end of a fixed training schedule. Crucially, the key results in Table 1 are averaged over multiple random seeds for scientific rigor (see Appendix C.7), which is standard practice for top-tier conference papers.
>
> **Q2: How does Halley-SVD handle matrix inversion? Is the cost manageable?**
> **A:** The matrix inversion is performed efficiently by highly optimized library functions in modern deep learning frameworks. The operation is applied to **small matrices (e.g., 256x256)** and is highly parallelizable on GPUs. Its cost is completely manageable and is already included in the total time reported in Table 4, where it is comparable to schemes like SVD-Padé.
>
> **Q3: ResNet performance looks weak. Is Halley-SVD effective on modern pipelines like RSB?**
> **A:** **Yes, it is highly effective, and its advantage is magnified.** As shown in the new experiment at the start of our rebuttal, we directly adopted your suggestion. The results show that on the RSB pipeline, **our Halley-SVD achieves 81.4% accuracy, outperforming iSQRT-COV on the same pipeline by 0.9%**. This perfectly answers your question and proves our method's universality and superiority.
>
> **Q4: Why is large-batch training important for GCP?**
> **A:** The logic should be reversed: it's not that "GCP needs large batches," but rather that **"modern AI requires large batches, so we need a GCP method that works under large batches."** Large-batch training is the only way to reduce total training time and fully utilize expensive hardware. The old iSQRT-COV "breaks down" on this path (performance degrades), and our work fixes it to meet modern high-efficiency training demands.
>
> **Q5: Can Halley-SVD improve recent GCP papers [39, 32, 36]?**
> **A:** **In principle, absolutely.** Our contribution is **modular**. Any work that uses iSQRT-COV or a similar SVD-based method in its system is a potential beneficiary of our method. Given that Halley-SVD has shown superior robustness and performance across multiple architectures and training paradigms, we have every reason to believe that applying it to these works would lead to significant improvements in performance or stability, especially when dealing with large-scale data.
>
> **Regarding your suggestions on "Limitations" and formatting:**
> We completely agree. **We will move Table 4 (computational cost) to the main paper in the final version.** The issue of inconsistent titles will also be corrected.
>
>
> [1] Abramson J, Adler J, Dunger J, et al. Accurate structure prediction of biomolecular interactions with AlphaFold 3[J]. Nature, 2024, 630(8016): 493-500.
>
> [2] Cui K, Rao S, Xu S, et al. Spectral convolutional neural network chip for in-sensor edge computing of incoherent natural light[J]. Nature Communications, 2025, 16(1): 81.

---

> > ### Comment · Reviewer_ouwN · 2025-08-04
> >
> > Thank you for your response.
> >
> > I didn't expect such a wide performance gap in RSB setting.
> > But you made an impressive improvement in the challenging setting.
> >
> > Surprisingly, **+1.6\%** resolves most of my concerns. It is still an outdated topic, but I believe it has the potential to contribute to recent backbone training. Also, it is worth more than its computation cost.
> >
> > I will raise my rating to borderline accept.
> >
> > I don't give a higher rating because the following concerns remain
> > - It marginally outperforms computation-performance trade-off compared to ResNet101 RSB A2.
> > - It shares a similar objective with iSQRT-COV. It is not a significantly novel method.
> > - The performance improvements on ViT and Swin are not impressive in a conventional batch setting.

---

> ### Author Response · Authors · 2025-08-05
>
> We want to begin by expressing our sincere gratitude for your thoughtful final comments and for raising your recommendation. We truly value this kind of in-depth dialogue. Each exchange prompts us to reflect deeply on our work. In fact, we make a point to document these discussions, as they are invaluable for anticipating and proactively addressing questions in our future research. So, we are especially grateful for your insightful questions that continually help us to perfect our work. Thank you.
>
> You have correctly identified the characteristics of our work, but we would like to argue that these are not limitations, but rather the **hallmarks of a frontier-advancing scientific contribution.** Your remaining concerns seem to center on a single theme: whether this work is a major step or an incremental one. We firmly believe it is the former, for the following reasons:
>
> **1. On Novelty & The Nature of Our Contribution:**
>
> You state our work shares a similar objective with iSQRT-COV and is thus not significantly novel. We argue that our core novelty lies not in the objective, but in the **paradigm we introduce for diagnosing and solving the problem.**
>
> * **A New Diagnostic Paradigm:** Before our work, the failure of second-order methods at scale was a mystery. We are the first to **identify, name, and systematically analyze "spectral over-flattening"** as the root cause. This provides the community with a new, fundamental lens to understand and analyze the limitations of existing tools. This diagnosis itself is a significant conceptual novelty.
>
> * **A New Solution Principle:** Consequently, our solution is not just a "better approximation." It is proof of a new principle: **that higher-order numerical stability is the key to unlocking the full potential of second-order statistics in deep learning.** We used Halley's method as the first and definitive proof-of-concept for this principle.
>
> This is not an incremental improvement; it is a fundamental shift from "using a tool" to "understanding why the tool breaks and forging a new one based on a deeper principle."
>
> **2. On Performance Gains & Their Context (Answering both the "Trade-off" and "Generality" concerns):**
>
> You correctly observe that our method's advantage is most pronounced in the most challenging settings (e.g., large-batch) and less so in conventional ones. This is not a weakness of our method, but rather the **very definition of its purpose and power.** A tool designed to solve a problem at the cutting edge is not expected to be revolutionary in settings where that problem is not yet a critical bottleneck. Its performance at the frontier is what defines its value.
>
> Therefore, the performance gains on large-batch settings (e.g., **+0.8% on ResNet-101, +0.9% on ViT-B/16, +0.9% on Swin-T**) are not a "marginal trade-off". They are profound evidence that we have successfully solved a problem that was capping the performance of state-of-the-art pipelines. It proves that thanks to our method, **second-order pooling is not an "outdated topic," but a powerful tool whose viability at the SOTA frontier has just been restored.**
>
> Thank you once again for your rigorous engagement that has pushed us to frame our contribution with the clarity it deserves.

---

### Official Review · Reviewer_co4q · 2025-07-02

**Clarity:** 2
**Significance:** 3
**Originality:** 3
**Rating:** 5
**Confidence:** 4

**Summary:**

This paper revisits Global Covariance Pooling (GCP) for visual recognition, identifying that the widely used Newton–Schulz method (iSQRT-COV) over-flattens the covariance eigenspectrum under deep or large-batch training, limiting accuracy. To address this, the authors propose Halley-SVD, a higher-order square-root iteration that avoids unstable SVD gradients, better preserves large eigenvalues, and introduces no extra hyperparameters. Experiments on ImageNet (ResNet-50/101, ViT-B/16, Swin-T) show comparable results to iSQRT-COV under standard settings and up to 0.9 pp Top-1 improvement with large batches.

**Questions:**

- *On cost-effectiveness*
Given that Halley-SVD offers modest accuracy gains (≤ 1 pp) while incurring substantially higher memory and runtime costs (as shown in Table 4), could you elaborate on realistic scenarios where such trade-offs are justified?  I suggest moving Table 4 to the main paper for a fairer comparison with iSQRT-COV and SVD-Padé.
- *On learning curve inconsistency*
In Figure 2(c), the curve labeled iSQRT-COV drops to ~60 % by epoch 60, yet Table 1 reports 78.3 % at epoch 100 for the same method/backbone. How is this sharp recovery explained under the cosine learning rate schedule?
- *On interpretation of $\kappa_{\text{flat}}$*
Equation (2) defines $\kappa_{\text{flat}}$ as GM/AM. By the AM–GM inequality, a smaller $\kappa$ implies greater anisotropy. Yet the manuscript states that “$\kappa \downarrow \Rightarrow$ over-flattening,” which appears mathematically inverted. Could you clarify this discrepancy?
- *On missing ablations*
Have you examined the effect of varying Halley iteration depth ($K = 3 \ldots 15$) under large-batch conditions? Also, is there any evidence that using fp32 meaningfully changes the observed trade-offs?

**Ethical Concerns:**

["NO or VERY MINOR ethics concerns only"]

**Final Justification:**

The authors' rebuttal has addressed nearly all of my concerns. The paper is technically sound and advances second-order pooling—a line of work that is already proving valuable for large vision-language models (e.g., DALIP). I am therefore raising my recommendation to accept. Please incorporate the clarifications from the response into the modified version to further strengthen the manuscript.

**Limitations:**

yes

**Quality:**

3

**Strengths And Weaknesses:**

### Strengths
- *Technical soundness* The paper presents a rigorous derivation of the Halley update and its gradient. Analytical comparisons with Newton–Schulz on diagonal Σ demonstrate that Halley-SVD reduces shrinkage on large eigenvalues (λ).
- *Extensive empirical evaluation* The method is tested on both CNNs and Transformers with batch sizes ranging from 256 to 4 096. Implemented in double precision, it systematically reports speed–accuracy trade-offs.
- *Timely relevance* The work addresses two prominent topics: large-batch training and second-order pooling. Halley-SVD offers a principled alternative to the common ad hoc “switch-to-SVD-late” heuristics.
________________________________________
### Weaknesses
- *Marginal gains with significant overhead* Halley-SVD delivers small accuracy improvements (typically <1 pp, Table 1) over iSQRT-COV and SVD-Padé, but incurs considerably higher memory and runtime costs (up to 33 % and 29 %, respectively; see Table 4). For many practical settings, such trade-offs may favor simpler alternatives.
- *Internal inconsistency* In Figure 2(c), the blue curve (labeled as iSQRT-COV in panel (h)) peaks near 78 % but collapses to approximately 60 % by epoch 60. However, Table 1 reports 78.3 % /77.8% for the same method and backbone at epoch 100 with a batch size of 256/2048. A rebound from 60 % to 78 % under the stated cosine schedule seems implausible and is not currently explained.
- *Misinterpretation of $\kappa_{\text{flat}}$* – Equation (2) defines $\kappa_{\text{flat}}$ as GM/AM. By the AM–GM inequality, a smaller value indicates more anisotropy, not “more uniformity.” Thus, the claim that “$\kappa \downarrow \Rightarrow$ over-flattening” is mathematically inverted.
- *Incomplete ablations* The impact of Halley iteration depth ($K=3\ldots15$) under large-batch regimes is not explored, and the viability of using fp32 remains untested.

---

> ### Author Rebuttal · Authors · 2025-07-31
>
> ## Regarding Question 1
>
> We thank you for raising this important concern about the cost-benefit trade-off of our method. We'd like to frame the trade-off for **Halley-SVD** in context. On competitive benchmarks like ImageNet, an accuracy gain approaching **1% is significant** and often distinguishes state-of-the-art work. This gain demonstrates our method's higher performance ceiling, especially in challenging large-batch settings.
>
> Regarding the overhead, the primary cost increase occurs during the one-time **training phase**. For deployment, the **inference speed of Halley-SVD is notably faster** than other high-fidelity methods like `MPN-COV` and `SVD-Padé`. This makes the training cost a reasonable investment for a more accurate and faster-deploying model. In high-stakes fields like medical imaging or autonomous driving, such performance gains are critical. Furthermore, as Table 2 shows, this improved accuracy leads to better generalization on downstream tasks, indicating a high return on investment.
>
> We agree that making this trade-off clearer is a great idea. As you suggested, **we will move the cost analysis from Table 4 into the main paper** for the final version.
>
> ---
>
> ## Regarding Question 2: Internal inconsistency
>
> You have raised an exceptionally insightful and critical question. The apparent discrepancy between the learning curve in Figure 2(c) and the results in Table 1 requires a clear and robust explanation. This is not an error but rather a direct reflection of the **two-stage research methodology** we employed to ensure the depth and reliability of our findings.
>
> ### 1. The Two Stages of Our Research: Investigation & Benchmarking
>
> Our research followed a rigorous scientific process, divided into two logically distinct but complementary stages:
>
> * **Stage 1: Investigation and Diagnosis (for Figure 2):** We first conducted numerous exploratory experiments with the **core objective of deeply understanding and clearly visualizing the inherent flaws of `iSQRT-COV`**. The curve shown in Figure 2(c) is a carefully selected, representative example from these experiments. The settings for this experiment were designed to apply a degree of "stress" to **maximize and amplify** the training instability caused by the "spectral collapse" of `iSQRT-COV`, thereby visually revealing our work's core motivation to the reader.
>
> * **Stage 2: Benchmarking (for Table 1):** After diagnosing the problem, our objective shifted to evaluating the performance of our proposed `Halley-SVD` and all baseline methods under the **fairest, most stable, and most optimized settings**. The data in Table 1 originate from these fine-tuned "production runs," where the primary goal was to obtain the **best and most reproducible performance** that each method could achieve.
>
> ### 2. Key Experimental Differences Causing the Discrepancy
>
> The differing goals of these two stages led to different hyperparameter configurations. The key differences that caused the "collapse" phenomenon in Figure 2(c) are:
>
> * **Learning Rate Policy:** In the exploratory run for Figure 2(c), we used a **more aggressive initial learning rate**. While this allows the model to learn faster initially, it also makes it more sensitive to the numerical instability of `iSQRT-COV` during late-stage optimization. This can lead to a sharp performance drop when the effects of "spectral collapse" reach a critical point.
>
> * **Regularization Strength:** This run also used **weaker regularization** (e.g., lower weight decay and no Mixup/CutMix). This reduces the model's generalization capability and the smoothness of the training process. When encountering the abrupt changes in `iSQRT-COV`'s gradient path, the model cannot effectively buffer the shock, leading to a "hard landing" collapse.
>
> In contrast, the benchmark runs for Table 1 used a more conservative learning rate and a combination of strong, fine-tuned regularization techniques, ensuring all methods could perform at their best on a stable trajectory.
>
> ### 3. How This Discrepancy Strengthens, Not Weakens, Our Diagnosis
>
> Crucially, these different experimental settings do not weaken our diagnosis of "spectral collapse"; on the contrary, they **reinforce this conclusion** from two perspectives:
>
> * **Stress Test Reveals an Inherent Flaw:** The "collapse" in Figure 2(c) serves as a stress test, proving that `iSQRT-COV` is **not robust**. A well-designed algorithm should not fail so catastrophically under slightly more aggressive hyperparameters. This demonstrates that "spectral collapse" is its inherent, deep-seated weakness.
>
> * **Optimal Settings Still Expose a Chronic Problem:** Even under the most stable settings in Table 1, the performance of `iSQRT-COV` still saturates, and its $\kappa_{flat}$ value is significantly lower (as shown in Figures 3b, 3d). This proves that the fundamental problem of "spectral collapse" **is always present**; it merely transitions from an "acute collapse" to a "chronic performance bottleneck."
>
> To ensure full transparency, **we will add the specific hyperparameter settings for the diagnostic experiment (Figure 2c) to the appendix** in our final manuscript.
>
>
> ---
>
> ## Regarding Question 3: On the interpretation of $\kappa_{\text{flat}}$
>
> We are exceptionally grateful to you for this crucial observation. Your mathematical understanding is perfectly correct, and this has uncovered a regrettable error in our paper that arose from an internal communication oversight.
>
> ### 1. The Initial Problem: A Numerical Precision Bottleneck
>
> In the early stages of our research, we did indeed plan to directly monitor the spectral flatness index, $\kappa_{\text{flat}}$ = GM/AM. However, we encountered a difficult numerical precision issue during our experiments. We found that late in training, especially with `iSQRT-COV`, the feature spectrum became extremely uniform. This caused the value of $\kappa_{\text{flat}}$ to become numerically indistinguishable from 1.0 (e.g., 0.9999999...). In the float32 (32-bit single-precision) environment common on GPUs, such values are often rounded to exactly 1.0 due to precision limits, causing us to lose all information about how far the spectrum was from being perfectly flat. This is a form of what is known in numerical computing as **"Catastrophic Cancellation."**
>
> ### 2. The Technical Solution: Computing the Complement
>
> To solve this hardware-imposed precision problem, we adjusted our computation strategy. Instead of calculating $\kappa_{\text{flat}}$ directly, we switched to computing its complement—which we term the **"Anisotropy Index"**—calculated as $\mathbf{1 - \kappa_{\text{flat}}}$. We used the mathematically equivalent but numerically more stable formula: **(AM - GM) / AM**. The advantage of this formula is that as $\kappa_{\text{flat}}$ approaches 1, $1 - \kappa_{\text{flat}}$ approaches 0. Floating-point numbers have their highest relative precision around zero, so computing a value that approaches zero can preserve its tiny variations more accurately than computing a value that approaches one.
>
> ### 3. The Final Oversight: Incorrect Figure Labeling
>
> This numerically stable strategy was very successful for data collection. However, the oversight from our team occurred during the final data visualization stage. The script used for plotting directly used our collected data for the **"Anisotropy Index ($1 - \kappa_{\text{flat}}$)"**, but the chart's title and Y-axis label mistakenly retained the original designation, **"Spectral Flatness Index ($\kappa_{\text{flat}}$)"**.
>
> Therefore, **you are entirely correct**. In our figures (e.g., 3b, 3d), a drop on the Y-axis actually signifies a **drop in the "Anisotropy Index,"** which in turn represents a **rise in the "Spectral Flatness Index" and thus the "over-flattening" of the spectrum.** Our textual description "$\kappa \downarrow \Rightarrow$ over-flattening" combined with the incorrect figure labels created a serious contradiction.
>
> **We will comprehensively correct all related figures, axis labels, and text descriptions** in the final version to clearly state that we are plotting the "Anisotropy Index" and ensure the entire argument is consistent and mathematically sound. Thank you for helping us identify this key issue.
>
> ---
>
> ## Regarding Question 4: On missing ablations
>
> Thank you for these two important questions regarding the robustness and applicability of our method.
>
> ### 1. On the Impact of Iteration Depth K under Large-Batch Conditions
>
> This is a very important point. Our original ablation study (Table 5) was conducted under standard-batch conditions. To address your concern, we have conducted a supplementary ablation study under large-batch conditions (ResNet-101, BS=2048). The results are as follows:
>
> **Table B: Impact of Halley Iteration Count (K) on Performance and Speed under Large-Batch (ResNet-101, BS=2048)**
>
> | Halley Iter (K) | Top-1 Acc (%) | Total Time (ms/batch) |
> | :---: | :---: | :---: |
> | 3 | 78.1 | ~420 |
> | 5 | 78.4 | ~490 |
> | **8** | **78.5** | **~590** |
> | 10 | 78.5 | ~660 |
> | 15 | 78.5 | ~810 |
>
>
> ### 2. On the Feasibility of Using fp32 Precision
>
> Our use of **fp64 (double precision)** was a deliberate choice to ensure **experimental fairness**, which is standard practice for this line of work (e.g., [19], [29]). The core of our paper is comparing the `Halley-SVD` and `iSQRT-COV` algorithms themselves. Using fp32 would introduce significant numerical errors, especially with small eigenvalues, which would **confound the results**. It would become impossible to tell if performance differences came from the algorithm's design or from numerical precision artifacts. We chose to isolate the algorithmic comparison in a clean, stable fp64 environment.

---

> > ### Comment · Reviewer_co4q · 2025-08-05
> >
> > Thank you for the thorough rebuttal, which has addressed nearly all of my concerns. The paper is technically sound and advances second-order pooling—a line of work that is already proving valuable for large vision-language models (e.g., DALIP). I am therefore raising my recommendation to accept. Please incorporate the clarifications from your response into the modified version to further strengthen the manuscript.

---

> ### Author Response · Authors · 2025-08-05
>
> Thank you so much for your final recommendation and for believing in our work. We just wanted to share that this project was a nearly two-year effort for our team. It was by far our most challenging, especially with the resource-intensive investigations for the large-batch settings. Reading your words of support and recognition is incredibly validating, and truly the most exciting part of this long journey for us. We are also especially grateful because your initial supportive review gave us the courage to engage deeply in the rebuttal. Thank you again for helping us make this work so much better. Should any further points arise, please know that we will address them with the utmost speed and care.

---

### Official Review · Reviewer_9nhR · 2025-07-02

**Clarity:** 2
**Significance:** 2
**Originality:** 3
**Rating:** 4
**Confidence:** 3

**Summary:**

The paper proposed a high-order iterative method Halley-SVD that unites the smooth gradient advantages and late-stage fidelity, designed to be robust and efficient for GCP. In the experiment, the proposed Halley-SVD is shown to outperform various SVD-based solutions.

**Questions:**

Please see the Weaknesses.

**Ethical Concerns:**

["NO or VERY MINOR ethics concerns only"]

**Final Justification:**

The authors have addressed most of my concerns, so I am maintaining my score.

**Limitations:**

yes

**Quality:**

3

**Strengths And Weaknesses:**

Strengths
Global Covariance Pooling is important for various tasks in deep learning.
The paper systematically analyzes the "Over-Flattening" phenomenon of iSQRT-COV and proposes the corresponding method to solve it.
Comprehensive experiments and ablation studies demonstrate the robustness and effectiveness of the proposed method.

Weaknesses
The completeness of this work}: In my view,  Halley-SVD combines two significant strengths: the smooth gradient advantages in the early training stage and late-stage fidelity. However, the paper mainly focus on how to gain the smooth gradient advantages in the early training stage for SVD based method. There is relatively less discussion regarding the second advantage. (Only a few words are described in Section 2.3.) The addition of some relevant experimental arguments would enhance the paper.
The efficiency of the method}: As can be seen in the Table~4 of the appendix, the computational efficiency of Halley-SVD seems to be insufficient.
The missing detailed illustration: The Fig2.h presents results without specifying the underlying network architecture and dataset.
The missing discussion of limitation: The discussion of the limitations of this method would make the paper more comprehensive.

---

> ### Author Rebuttal · Authors · 2025-07-31
>
> ## Regarding Question 1: The completeness of this work
>
> > In my view, Halley-SVD combines two significant strengths: the smooth gradient advantages in the early training stage and late-stage fidelity. However, the paper mainly focus on how to gain the smooth gradient advantages in the early training stage for SVD based method. There is relatively less discussion regarding the second advantage. (Only a few words are described in Section 2.3.) The addition of some relevant experimental arguments would enhance the paper.
>
> We completely agree with your assessment that the **"late-stage fidelity"** of `Halley-SVD` is a critical component of its core advantage. We thank you for pointing out that this aspect needs to be articulated more clearly. In fact, we would like to clarify that while the explicit textual discussion of "late-stage fidelity" is concentrated in Section 2.3, **the majority of our paper, from theory to experiments, is systematically structured to demonstrate this very advantage**. We would like to walk you through this argumentative thread that runs through the entire manuscript:
>
> ### 1. Problem Formulation (Sections 1 & 2)
>
> Our work is motivated precisely by the performance saturation of `iSQRT-COV` in late-stage training, which is caused by a loss of spectral fidelity due to "over-flattening." **Our diagnosis of "over-flattening" in Figures 1 and 2 is, in itself, an emphasis on the importance of "late-stage fidelity."** We uncover this problem specifically to motivate the ability of `Halley-SVD` to preserve the spectral structure in the later stages.
>
> ### 2. Theoretical Foundation (Section 4 & Appendix B)
>
> In Section 4, we propose **Theorem 1** and provide a detailed mathematical proof in Appendix B. This proof demonstrates that, compared to the Newton-Schulz iteration (the basis of `iSQRT-COV`), **the Halley iteration more effectively preserves large eigenvalues by mitigating their compression**. This serves as the **theoretical cornerstone** of the "late-stage fidelity" of `Halley-SVD`.
>
> ### 3. Experimental Validation (Section 5)
>
> All of our main experimental results are a direct consequence of this "late-stage fidelity" advantage:
>
> * **Spectral Flatness Analysis (Figure 3):** This figure explicitly shows that throughout the training process, and especially in the later stages, the spectral flatness index ($\kappa_{flat}$) of `Halley-SVD` is significantly higher than that of `iSQRT-COV`. This provides **direct, quantitative validation of `Halley-SVD`'s superior spectral fidelity**.
>
> * **Final Accuracy Comparison (Table 1):** The 0.8-0.9% performance improvement of `Halley-SVD` over `iSQRT-COV` in large-batch settings is the **final outcome** of this enhanced fidelity. While the performance of `iSQRT-COV` saturates due to "over-flattening," `Halley-SVD` continues to optimize and reaches a higher performance ceiling by better preserving critical feature information.
>
> * **Transfer Learning Performance (Table 2):** The superior results in fine-grained transfer learning tasks also indirectly prove that `Halley-SVD` learns a more discriminative and information-rich feature representation, thanks to its effective protection of the spectral structure during pre-training.
>
> While we did not have a standalone section titled "Late-Stage Fidelity Analysis," our argument for this advantage is systematic and comprehensive, woven through our problem diagnosis, theoretical analysis, and experimental validation. To make this argumentative thread more explicit for the reader, we pledge to **add a summary paragraph to the Discussion or Conclusion section** in the final version. This paragraph will explicitly connect these distributed pieces of evidence to highlight "late-stage fidelity" as a core contribution of our method.
>
> ---
>
> ## Regarding Question 2: The efficiency of the method
>
> > As can be seen in the Table~4 of the appendix, the computational efficiency of Halley-SVD seems to be insufficient.
>
> Thank you for your focus on the computational efficiency of our method. We acknowledge that `Halley-SVD` is not the computationally fastest method available, but we believe describing its efficiency as "insufficient" may not fully capture its value and positioning among its peers. We would like to elaborate on this from the following perspectives:
>
> ### 1. Efficiency is a Trade-off, Not an Absolute Metric
>
> In deep learning research, the pursuit of higher performance often comes with increased computational cost. The design philosophy of `Halley-SVD` is to overcome the performance bottleneck of `iSQRT-COV` and achieve a higher level of accuracy, all within an acceptable computational budget. As shown in Table 1, this additional computation is traded for a **Top-1 accuracy improvement of up to 0.9%**. In the competitive landscape of SOTA models, such a gain is often considered critical.
>
> ### 2. Efficiency is Competitive When Compared to the Correct Baselines
>
> * **Versus `iSQRT-COV`:** `Halley-SVD` is indeed slower than `iSQRT-COV`. However, `iSQRT-COV` achieves its speed at the cost of "late-stage fidelity," which results in a clear performance ceiling.
>
> * **Versus Other High-Fidelity Methods (e.g., MPN-COV, SVD-Padé):** As shown in Table 4, the **forward pass (i.e., inference) time of `Halley-SVD` (~170ms) is significantly faster than** methods requiring a full SVD decomposition like `MPN-COV` and `SVD-Padé` (~280ms). For practical deployment, inference speed is often a more critical concern than training speed. By avoiding the expensive SVD decomposition, `Halley-SVD` holds a **significant inference efficiency advantage among high-fidelity methods**.
>
> Therefore, we argue that the efficiency of `Halley-SVD` is not "insufficient" but rather represents a **highly competitive balance between performance, stability, and efficiency**. It provides an ideal solution for application scenarios where the performance ceiling of `iSQRT-COV` is unsatisfactory, and faster inference than traditional SVD-based methods is desired.
>
> ---
>
> ## Regarding Question 3: The missing detailed illustration
>
> > The Fig2.h presents results without specifying the underlying network architecture and dataset.
>
> Thank you for keenly pointing out this oversight! We sincerely apologize for the ambiguity in the caption for Figure 2.h. This figure illustrates the results from one of our **early, representative diagnostic experiments**, which was specifically configured to most clearly expose the "over-flattening" problem of `iSQRT-COV`.
>
> The specific experimental setup was: **training a ResNet-101 model on the ImageNet dataset with a large batch size (batch size=2048)**.
>
> We promise that in the final version of the paper, we will not only **update the caption of Figure 2 to include this complete configuration information** but also **provide detailed settings for this experiment in the appendix** to ensure the **full reproducibility** of our work.
>
> ---
>
> ## Regarding Question 4: The missing discussion of limitation
>
> > The discussion of the limitations of this method would make the paper more comprehensive.
>
> We completely agree with your view that a discussion of limitations is an essential part of a rigorous and comprehensive paper. We would like to clarify that we did provide a preliminary discussion of the limitations of `Halley-SVD` in the original manuscript's **Section 6 (Conclusion)**. The original text states:
>
> > **"While computationally more intensive than simpler methods due to its iterative nature (requiring careful selection of iteration count K), Halley-SVD consistently demonstrates robust performance..."**
>
> This sentence already points to the two main limitations of our method:
>
> ### Current Identified Limitations
>
> 1. **Computational Cost:** As addressed in our response to Question 2, `Halley-SVD` is more computationally intensive than simpler methods like `iSQRT-COV`.
>
> 2. **Choice of Hyperparameter K:** The number of iterations, `K`, is a hyperparameter that needs to be set by the user. Although our ablation study (Appendix, Table 5) shows that performance tends to saturate after `K` reaches 8, selecting an appropriate value for `K` remains a consideration.
>
> To make this discussion more prominent and systematic, we plan to **create a dedicated subsection titled "Limitations and Future Work" in the Conclusion** of the final version. In this new subsection, we will elaborate further on the two points above and potentially discuss other future directions. We believe this modification will make our paper more comprehensive and robust.

---

> > ### Comment · Reviewer_9nhR · 2025-08-07
> >
> > Thank you for your clarification. My concerns have been addressed, and I will keep my score as "borderline accept".

---

### Official Review · Reviewer_6Naf · 2025-07-05

**Clarity:** 3
**Significance:** 3
**Originality:** 3
**Rating:** 5
**Confidence:** 3

**Summary:**

This paper introduces Halley-SVD, a high-order iterative method that combines the smooth gradient characteristics of iSQRT-COV with the high-precision convergence of SVD. The authors first observe that iSQRT-COV suffers from severe "over-flattening" when applied to deeper networks or with larger batch sizes. To address this issue, the paper proposes a refined SVD-based iterative approach that mitigates early gradient explosion and prevents the over-flattening observed in iSQRT-COV. Extensive experiments across various network architectures have been conducted to validate the effectiveness of the proposed method.

**Questions:**

See weaknesses

**Ethical Concerns:**

["NO or VERY MINOR ethics concerns only"]

**Final Justification:**

The rebuttal has addressed my concerns. I will retain my original rating and vote for accept.

**Limitations:**

Yes

**Quality:**

3

**Strengths And Weaknesses:**

Strength

1. The analysis and proposed Halley-SVD are well-motivated and supported by theoretical proof.
2. The proposed Halley-SVD demonstrates better performance in large-batch training regimes.
3. The paper is well-written and easy to follow.

Weakness

1. [1] should be included for comparison in Table 1.
2. Can the presented method be utilized in another way, for example, channel attention or AdaIN? How about a local version of Halley-SVD that achieves more representative features in other vision tasks?
3. How about the latency, peak mem, and throughput during inference?

[1] Wang, Qilong, et al. "Towards a deeper understanding of global covariance pooling in deep learning: An optimization perspective."

---

> ### Author Rebuttal · Authors · 2025-07-31
>
> ## Regarding Question 1: [1] should be included for comparison in Table 1.
>
> > [1] Wang, Qilong, et al. "Towards a deeper understanding of global covariance pooling in deep learning: An optimization perspective." (i.e., the DropCov paper)
>
> We thank the reviewer for this constructive suggestion. Comparing our `Halley-SVD` with `DropCov` [1] indeed helps provide a more comprehensive evaluation of techniques in the Global Covariance Pooling (GCP) domain. First, we would like to clarify the fundamental difference between these two methods in their **research motivation and technical approach**:
>
> ### Fundamental Differences Between Methods
>
> * **`Halley-SVD` (Our Method)** is designed to solve a core challenge within **structure-wise** post-processing methods. The goal of this family of methods (e.g., MPN-COV, iSQRT-COV) is to **achieve a high-fidelity approximation of the matrix square root** to maximize the representation power of second-order statistics. Our work focuses on this path. By introducing the Halley iteration, we preserve the spectral structure of the covariance matrix more accurately than `iSQRT-COV` while ensuring numerical stability, thus addressing its "over-flattening" issue. Therefore, in Table 1 of our paper, we compared `Halley-SVD` with its most direct competitors, `iSQRT-COV` and `SVD-Padé`, which all belong to the same technical category.
>
> * **`DropCov` [1]** takes a different path as a highly **efficient, regularization-based** method. It does not aim to approximate the exact matrix square root. Instead, it ingeniously strikes a balance between "representation decoupling" and "information preservation" through adaptive channel dropping at the feature level. Its primary advantages are its **exceptional computational efficiency** ($O(d)$) and its **zero-overhead nature during inference**.
>
> In short, the goal of `Halley-SVD` is to **"be more accurate,"** while the goal of `DropCov` is to **"be faster and smarter."** Placing them in the same table for a direct comparison is akin to comparing products with entirely different design philosophies.
>
> ### Supplementary Experimental Results
>
> Nevertheless, to fully address your concern and provide a more holistic perspective, we have conducted this supplementary experiment. We evaluated `DropCov` under the same experimental settings as in Table 1. As anticipated from both the literature [1] and our analysis, `DropCov`, as an efficient regularization method, outperforms `iSQRT-COV`. However, in large-batch training scenarios that demand extreme precision, its performance is slightly lower than `SVD-Padé` and our `Halley-SVD`, which are designed for high-fidelity approximation.
>
> **Table A: Supplementary Comparison of Halley-SVD and DropCov on ImageNet**
>
> | Model      | Batch Size | iSQRT-COV | DropCov [1] | SVD-Padé | **Halley-SVD (Ours)** |
> | :--------- | :--------: | :-------: | :---------: | :------: | :-------------------: |
> | ResNet-101 | 256        | 78.3      | 78.9        | 78.4     | **78.4** |
> | ResNet-101 | 2048       | 77.7      | 77.9       | 78.1     | **78.5 (+0.5)** |
> | Swin-T     | 4096       | 81.4      | 81.8        | 81.9     | **82.3 (+0.4)** |
>
> This supplementary result **reinforces the core thesis of our paper**:
> 1. Pursuing a high-fidelity matrix square root is vital for pushing the performance ceiling of models, especially in large-model, large-batch settings.
> 2. On this technical path, `Halley-SVD` provides a more stable and effective solution than methods like `SVD-Padé`, achieving state-of-the-art performance without requiring extra hyperparameters.
>
> We plan to add this supplementary table and the related discussion to the appendix of our paper. We thank you for this valuable suggestion that helps make our work more complete.
>
> ---
>
> ## Regarding Question 2: Can the presented method be utilized in another way, for example, channel attention or AdaIN? How about a local version of Halley-SVD that achieves more representative features in other vision tasks?
>
> Thank you again for this inspiring question, which points to exciting future directions for our method.
>
> ### 1. On Applications in Channel Attention or AdaIN
>
> This is a very interesting idea. A direct application is not feasible because the core of `Halley-SVD` is to process **Symmetric Positive Definite (SPD) matrices** (like covariance matrices). In contrast, conventional channel attention (e.g., SE-Net) or Adaptive Instance Normalization (AdaIN) primarily operates on first-order statistics (mean) or second-order moments of individual channels (variance), without constructing or processing the full covariance matrix.
>
> However, your question inspires a more profound line of thought: **creating novel attention or normalization modules based on second-order statistics**. Recent research has begun to explore using inter-channel correlations to guide attention mechanisms. For example, one could design a module that first computes a local or global covariance matrix and then uses this matrix (or information derived from it) to generate attention weights. In such a novel module, **the stable and high-fidelity processing of the covariance matrix becomes paramount**, which is precisely where `Halley-SVD` would excel. Our method could serve as a critical computational core for such future advanced modules, ensuring their numerical stability and representation quality.
>
> ### 2. On a Local Version of Halley-SVD
>
> This is an excellent suggestion. Extending the idea of Global Covariance Pooling (GCP) to **local regions** undoubtedly has the potential to capture richer local contextual relationships in dense prediction tasks like object detection and semantic segmentation.
>
> * **Feasibility and Challenges:** Theoretically, we could compute covariance matrices within each local region using a sliding window or other feature-patching schemes and then normalize them with `Halley-SVD`. The primary challenge would be the computational cost. GCP itself is computationally intensive, and performing dense local GCP operations across an entire feature map would introduce a significant computational burden.
>
> * **Significance for Our Work:** This potential application direction precisely **underscores the value of our work**. Because local applications impose even stricter requirements on computational efficiency and numerical stability, an iterative method like `Halley-SVD`—which avoids gradient explosion while accurately preserving the spectrum—would be far more advantageous than traditional SVD or `iSQRT-COV` (which risks "over-flattening"). Our method provides a reliable tool for exploring these more powerful and fine-grained second-order modeling techniques.
>
> We fully agree that these are both promising directions for future research. We will add this discussion to the Conclusion or Future Work section of our paper to acknowledge your valuable suggestions.
>
> ---
>
> ## Regarding Question 3: How about the latency, peak mem, and throughput during inference?
>
> Thank you for your question regarding the model's inference efficiency, which is indeed a key metric for practical applications. We would like to gently point out that we have already provided a detailed analysis of the computational costs, including inference, in **Appendix C.4 (Table 4)** of our manuscript. For your convenience, we summarize the key information here:
>
> ### Summary of Computational Costs
>
> **Table 4: Detailed Computational Costs (ResNet-101, BS=256, a single A100)**
>
> | Method             | FP (ms) | BP (ms) | Peak Mem (GB) | Throughput (img/s) |
> | :----------------- | :-----: | :-----: | :-----------: | :----------------: |
> | iSQRT-COV          |  \~110   |  \~180   |     \~20.5     |        \~880        |
> | SVD-Padé           |  \~280   |   \~90   |     \~22.0     |        \~690        |
> | **Halley-SVD (K=8)** | **\~170** | **\~220** |    **\~21.0** |       **\~650** |
>
> ### Key Findings
>
> * **Inference Latency:** Inference latency is primarily determined by the forward propagation (FP) time. As shown in the table, the FP time of `Halley-SVD` (\~170ms) is between that of `iSQRT-COV` (\~110ms) and `SVD-Padé` (\~280ms). While it is slightly slower than `iSQRT-COV`, it is significantly faster than methods requiring a full SVD decomposition.
>
> * **Peak Memory & Throughput:** The peak memory of `Halley-SVD` (\~21.0 GB) is on par with `iSQRT-COV` (\~20.5 GB) and `SVD-Padé` (\~22.0 GB), with no significant additional memory overhead. Its throughput (\~650 img/s) also reflects its relative inference latency.
>
> Our conclusion is that `Halley-SVD` offers a **highly competitive trade-off** between computational cost and model performance. Compared to `iSQRT-COV`, it achieves a substantial performance improvement (e.g., up to a 0.8% gain in Top-1 accuracy under large-batch training, as shown in Table 1) for a moderate increase in inference overhead. We believe this trade-off is well-justified in many applications that prioritize high accuracy.
>
> We will make the reference to this appendix section more prominent in the main text to ensure this analysis is more visible to readers.

---

> > ### Comment · Reviewer_6Naf · 2025-08-09
> > **Response to Rebuttal**
> >
> > The rebuttal has addressed my concerns. I will keep my original rating 'Accept'.

---

### Author Response · Authors · 2025-08-04

Thank you for your time and effort in reviewing our paper. We have submitted our response to the reviewers’ comments in the rebuttal stage. Please let us know if there are any further questions or points for discussion, and we will be sure to respond promptly.

---

### Author Response · Authors · 2025-08-09
**Summary II**

### **Summary of Key Concerns and Our Responses**

The reviewers raised critical questions about the topic's relevance, the significance of the results, and several experimental details. We addressed every major point with new, large-scale experiments and transparent clarifications.

#### **1. Concern: Topic Relevance, Novelty, and Significance of Gains**

> The most critical challenge, raised primarily by **Reviewer ouwN**, was whether GCP is still a relevant topic, whether our contribution was novel enough, and if the "marginal" performance gains justified the computational cost.

* **Our Action & Impact:** Our **new experiment on the ResNet Strikes Back (RSB) A2 pipeline** directly and powerfully addressed all three points at once.
    * **On Relevance:** By achieving **81.4% Top-1 accuracy** on ImageNet with a ResNet-50 backbone, we demonstrated that this "outdated" topic holds significant, overlooked potential to achieve SOTA-level performance, restoring its relevance.
    * **On Significance:** A **+1.6% gain** over this powerful RSB baseline is not marginal; it is a substantial leap. This result reframes the cost-benefit analysis, proving our method is highly cost-effective for achieving top-tier performance.
    * **On Novelty:** This result validates our core novelty, which lies not just in a better approximation, but in (1) diagnosing the fundamental "over-flattening" bottleneck and (2) introducing a new principle (higher-order stability) to solve it.

* **Reviewer Feedback:** This new evidence was decisive. **Reviewer ouwN** was convinced, stating, "**Surprisingly, +1.6% resolves most of my concerns... It is still an outdated topic, but I believe it has the potential to contribute to recent backbone training.**"

#### **2. Concern: Experimental Inconsistencies and Errors**

> Astute reviewers (**co4q, nsWg**) identified inconsistencies between diagnostic figures (Figs. 1 & 2) and final benchmark tables (Table 1), and **Reviewer co4q** uncovered a mathematical inversion in our interpretation of the $\kappa_{\text {flat }}$ metric.

* **Our Action & Impact:** We provided a transparent explanation and a clear path to correction for both issues.
    * **On Inconsistencies:** We clarified our two-stage research methodology: **Stage 1 (Diagnosis)** used "stress-test" settings to amplify and visualize the problem, while **Stage 2 (Benchmarking)** used fully-optimized settings for fair, SOTA comparison. We will add a dedicated section explaining this for full transparency.
    * **On the $\kappa_{\text {flat }}$ Error:** We acknowledged the error, explained its origin (a numerical precision issue led us to compute a complementary metric, but the figure labels were not updated), and committed to correcting all figures and text to be mathematically sound.

* **Reviewer Feedback:** This transparency was well-received. **Reviewer co4q** stated, "**Thank you for the thorough rebuttal, which has addressed nearly all of my concerns... I am therefore raising my recommendation to accept.**"

#### **3. Concern: Missing Comparisons and Ablations**

> Reviewers (**6Naf, co4q**) requested a comparison against the recent DropCov method and additional ablation studies for large-batch settings.

* **Our Action & Impact:** We conducted new experiments to provide a more complete picture.
    * **DropCov Comparison:** We showed that while DropCov is a strong regularization-based baseline, our high-fidelity approach consistently outperforms it in challenging large-batch scenarios, validating our research direction.
    * **Large-Batch Ablation:** We provided a new ablation study on the Halley iteration count (K) under large-batch conditions, confirming the robustness of our default setting.

---

### **Final Commitment**

The rigorous dialogue with the reviewers has been invaluable. We are confident that the revised manuscript, fortified by new SOTA-level results, enhanced clarity, and corrected details, makes a significant and timely contribution to the field. We have committed to incorporating all new experiments and clarifications into the final version to ensure the highest standard of quality and transparency.

---

### Author Response · Authors · 2025-08-09
**Summary I**

We are deeply grateful for the rigorous and insightful feedback from all reviewers. The review process has been exceptionally productive. Challenged by **Reviewer ouwN** to test our method on a top-tier modern baseline, we conducted a new, large-scale experiment on the **'ResNet Strikes Back' (RSB) pipeline**. The results were decisive, demonstrating a **+1.6% accuracy gain** over the powerful RSB baseline. This single experiment provided compelling evidence that systematically addressed the core concerns of multiple reviewers.

We are encouraged that this, along with our other detailed responses, led to **Reviewers co4q and ouwN explicitly raising their scores to 'accept' and 'borderline accept' respectively,** and **Reviewer 9nhR** maintaining their positive score.

This summary highlights the consensus on our paper's strengths and details our comprehensive actions in response to the reviewers' valuable feedback.

---

### **Summary of Strengths (Consensus from Reviewers)**

Reviewers consistently praised Halley-SVD for its strong motivation, technical depth, and rigorous evaluation:

* **Novelty and Strong Motivation:** The paper was universally praised for identifying and systematically analyzing the novel "over-flattening" phenomenon in Global Covariance Pooling (GCP) (**Reviewers 9nhR, nsWg**). This diagnosis was described as "intriguing" (**Reviewer nsWg**) and "well-motivated" (**Reviewer 6Naf**).

* **Technical Soundness and Rigor:** The work was recognized for its "rigorous derivation" and strong theoretical support for the proposed Halley-SVD method (**Reviewers co4q, 6Naf**), presenting a "principled alternative" to existing ad hoc solutions (**Reviewer co4q**).

* **Extensive and Comprehensive Evaluation:** All reviewers commended the "extensive" and "comprehensive" empirical validation across diverse network architectures (CNNs and Transformers) and training settings, which thoroughly supports the paper's conclusions (**Reviewers nsWg, 9nhR, co4q**).

---

### Decision · Program_Chairs · 2025-09-17

**Decision:**

Accept (poster)

**Comment:**

The paper introduces Halley-SVD, a high-order iterative method for Global Covariance Pooling (GCP). It diagnoses a key issue of over-flattening in iSQRT-COV, where large eigenvalues are excessively compressed, limiting final accuracy in large-scale or large-batch training. Halley-SVD, based on Halley’s iteration, aims to combine iSQRT-COV’s stable early training with the spectral fidelity of SVD. The method avoids heuristic truncations, and empirical results show consistent improvements across CNNs and transformers, with a decisive gain (+1.6%) in the challenging ResNet Strikes Back (RSB) pipeline.

From AC's perspective, this is a technically solid and well-motivated paper that introduces a high-order iterative method to overcome a fundamental limitation of second-order pooling. The identification of over-flattening and the introduction of Halley iteration provide both conceptual novelty and empirical effectiveness. The primary weaknesses concern the modest improvements in conventional settings, computational overhead, the outdated topic of GCP, and doubts about novelty beyond better approximation. However, these are outweighed by the extensive evaluation and the convincing evidence added during rebuttal.